



# Effect of latitudinally displaced gravity wave forcing in the lower stratosphere on the polar vortex stability

Nadja Samtleben[1], Christoph Jacobi[1], Petr Pišoft[3], Petr Šácha[2,3,4], and Aleš Kuchař[3]

[1]Institute for Meteorology, Universität Leipzig, Stephanstr. 3, 04103 Leipzig, Germany
[2]Institute for Meteorology, Universität für Bodenkultur Wien, Gregor-Mendel-Straße 33, 1180 Vienna, Austria
[3]Department of Atmospheric Physics, Faculty of Mathematics and Physics, Charles University, V Holesovickach 2, 180 00 Prague 8, Czech Republic
[4]EPhysLab, Faculty of Sciences, Universidade de Vigo, Campus As Lagoas, s/n, 32004 Ourense, Spain

**Correspondence:** Nadja Samtleben (Nadja.Samtleben@uni-leipzig.de)

**Abstract.** In order to investigate the impact of a locally confined gravity wave (GW) hotspot, a sensitivity study based on simulations of the middle atmosphere circulation during northern winter was performed with a nonlinear, mechanistic, global circulation model. To this end, for the hotspot region we selected a fixed longitude range in the East Asian region (120°E-170°E) and a latitude range from 22.5°N-52.5°N between 18 km and 30 km, which was then shifted northward in steps of 5°. For the southernmost hotspots, we observe a decreased stationary planetary wave (SPW) 1 activity in the upper stratosphere/lower mesosphere, i.e. less SPWs 1 are propagating upwards. These GW hotspots are leading to a negative refractive index inhibiting SPW propagation at midlatitudes. The decreased SPW 1 activity is connected with an increased zonal mean zonal wind at lower latitudes. This in turn decreases the meridional potential vorticity gradient ($q_y$) from midlatitudes towards the polar region. A reversed $q_y$ indicates local baroclinic instability which generates SPWs 1 in the polar region, where we observe a strong positive Eliassen-Palm (EP) divergence. Thus, the EP flux is increasing towards the polar stratosphere (corresponding to enhanced SPW 1 amplitudes) where the SPWs 1 are breaking and the zonal mean zonal wind is decreasing. Thus, the local GW forcing is leading to a displacement of the polar vortex towards lower latitudes. The effect of the local baroclinic instability indicated by the reversed $q_y$ also produces SPWs 1 in the lower mesosphere. The effect on the dynamics in the middle atmosphere by GW hotspots which are located northward of 50°N is negligible because the refractive index of the atmosphere is strongly negative in the polar region. Thus, any changes in the SPW activity due to the local GW forcing are quite ineffective.

## 1 Introduction

During winter the dynamics of the middle atmosphere are mainly dominated by the polar vortex, which develops due to the lack of incoming solar radiation, and which is modified by the impact of atmospheric waves with different spatial and temporal scales (Douville, 2009). The most important characteristic of atmospheric waves is their ability to transport and deposit energy and momentum. In particular, gravity waves (GWs) mainly developing in the troposphere distribute energy and momentum throughout the whole atmosphere thereby maintaining the circulation and the thermal structure of the upper atmosphere (Fritts



and Alexander, 2003). They also contribute to turbulence and mixing between all vertical layers. Their most important sources are orography (Smith, 1985; Nastrom and Fritts, 1992), convection (Tsuda et al., 1994), frontal systems (Plougonven and Zhang, 2014) or spontaneous adjustment processes (Fritts and Alexander, 2003). Strongly depending on the phase speed $c$ and the background wind $u$, GWs are able to propagate into the middle atmosphere. Due to the exponentially decreasing density

of the atmosphere the GW amplitude is exponentially increasing with height. Usually, the GW spectrum is already saturated in the stratosphere, which means that GW amplitudes cannot grow anymore and, according to the linear theory, partly break. This effect is the stronger the closer their phase speed $c$ is to the background wind $u$. If $c$ is equal to $u$, the GW encounters its critical line and cannot propagate anymore (Lindzen, 1981). Thus, mainly GWs propagating into the opposite direction than the background wind are usually observed in the middle atmosphere. Also GWs being faster than the background wind are able

to propagate but they are mostly filtered out by the strong polar-night jet. For this reason the wind reverses in the mesosphere due to GW breaking, while in the opposite direction to $u$ travelling GWs deposit their momentum (Lindzen, 1981; Holton, 1982). The transfer of energy and momentum by breaking GWs is also called GW drag.

Owing to the variety of sources GWs have a large spatial and temporal variability. To capture the global distribution of GWs, the potential energy ($E_{pot}$), momentum flux (MF), or stability indicators (Pišoft et al., 2018) can be estimated by using satellite

data (Ern et al., 2004; Fröhlich et al., 2007; Hoffmann et al., 2013; Schmidt et al., 2016). These numerous observational studies highlight a number of different local GW hotspots, which are mainly generated by orography and convection. The most common GW hotspots are the orographically induced GW hotspots near the Alps (Hierro et al., 2018), the Andes (Llamedo et al., 2009; Alexander et al., 2010; Lilienthal et al., 2017), the Antarctic Peninsula (Moffat-Griffin et al., 2010), the Himalayas (Kumar et al., 2012), the Mongolian Plateau (White et al., 2018), the Rocky Mountains (Lilly et al., 1982) and in the Scandinavian

region (Kirkwood et al., 2010), and convectionally generated GWs at the intertropical convergence zone (Dias and Pauluis, 2009), in India due to the monsoon, and in North America due to thunderstorms (Hoffmann and Alexander, 2010). However, objective determination of the GW drag from satellite measurements alone is not possible. Based on in-situ and ground-based measurements there are indications that the GWs are already breaking in the lower stratosphere (LS). E.g., Plougonven et al. (2008), having performed superpressure balloon measurements over the Antarctic Peninsula analysed the local potential tem-

perature gradients and the GW drag and figured out that the GWs are breaking in this region. Also Constantino et al. (2015), having analysed by thunderstorm generated GWs on the basis of lidar measurements, calculated directly the GW drag in the stratosphere.

This study is focusing on different GW breaking areas in the lower stratosphere along and the effects on the middle atmosphere highly depending on the position. It is motivated by the findings of Šácha et al. (2015) who were concentrating on the East

Asian/North Pacific (EA/NP) region near Japan where they observed a GW hotspot being active during equinoxes and winter solstices. The GWs are orographically and convectively generated due to the topography directly at the coastline and the warm Kuroshio current. Šácha et al. (2015) analysed the local instabilities by calculating the Richardson number and by analysing reanalysis data and found that the GWs are breaking in this area. Based on these results they simulated the observed Asian GW breaking hotspot with a global circulation model (GCM) and analysed its effect in the middle atmosphere circulation (Šácha

et al., 2016). According to previous publications e.g. by Smith (2003), Lieberman et al. (2013), or Matthias and Ern (2018),





Šácha et al. (2015) observed a forcing of additional stationary planetary waves (SPWs) due to a longitudinally variable GW drag. We are pursuing this idea by shifting meridionally the EA/NP hotspot along its fixed longitude range to get information about its impact on the middle atmosphere at different latitudinal positions. Therefore, the EA/NP GW hotspot is our starting point, from which we are displacing the GW hotspot towards lower and higher latitudes in 5° steps. In section 2 of this paper,

we provide a brief description of the GCM and detail the implementation of the GW hotspot within the GCM. In section 3 we describe and discuss the observed effects of the GW hotspots on the circulation of the middle atmosphere by analysing the stationary planetary wave activity and the propagation conditions. Finally, conclusions and outlook are presented in section 4.

## 2   Numerical model experiments

### 2.1   Model description and set up

To investigate the effect of localised GW breaking hotspots in the LS simulations have been performed using the Middle and Upper Atmosphere Model (MUAM, Pogoreltsev et al. (2007)). MUAM is a non-linear mechanistic 3D grid point model, which is an updated version of the global circulation model COMMA-LIM (Fröhlich et al., 2003a, 2007; Jacobi et al., 2006). The model extends in 56 layers up to an altitude of about $160\,km$ in logarithmic pressure height. In the lowermost $10\,km$, zonal mean temperatures are nudged to 2000-2010 mean monthly mean ERA Interim (Dee et al., 2011) zonal mean temperatures

to correct the climatology of the troposphere, which is not included in detail in the model (Jacobi et al., 2015; Lilienthal et al., 2018). Furthermore, at $1000\,hPa$, which defines the lower boundary of the model, SPWs of wavenumbers 1, 2 and 3 are forced, which are extracted from 2000-2010 mean ERA Interim monthly temperature and geopotential reanalysis data. The horizontal resolution of the model is 5° in latitude and 5.625° in longitude and the vertical resolution is $2.842\,km$ with a constant scale height of H = $7\,km$. The model solves the primitive equations in flux form (e.g. Jakobs et al., 1986). MUAM

includes parameterizations to simulate subgrid processes such as GWs, absorption of solar radiation, or infrared cooling. The absorption of radiation is realized according to Strobel (1986). This parameterization is focused on the absorption processes due to trace gases such as $H_2O$ (absorber in the troposphere), $CO_2$ and $O_3$ (absorber in the stratosphere). Water vapor and ozone fields are prescribed. The heating rates are calculated by absorption bands representing the wavelength interval, in which these trace gases are absorbing the atmospheric radiation. Infrared emission of $CO_2$ is parameterized after Fomichev et al. (1998),

and ozone infrared cooling in the $9.6\,\mu m$ band is calculated after Fomichev and Shved (1985).

GWs are parameterized after an updated linear scheme (Lindzen, 1981; Jakobs et al., 1986) with multiple breaking levels (Fröhlich et al., 2003b; Jacobi et al., 2006). GW amplitudes are included at an altitude of $10\,km$ as zonal mean with a global average of $1\,cms^{-1}$ for the vertical velocity perturbation. This value is weighted by a prescribed zonal mean GW amplitude distribution based on potential energy data obtained from GPS radio occultation measurements (Šácha et al., 2015; Lilienthal

et al., 2017). At each grid point 48 waves are induced propagating in eight different directions with six different phase speeds ranging from 5 to $30\,ms^{-1}$.

In this configuration based on January decadal mean (2000-2010) ERA Interim reanalysis data we create a reference simulation with a spin-up period of 270 days, in which the mean circulation is built up and different waves like planetary waves (PWs)




and tides are generated. The declination and the ozone and carbon dioxide concentration are fixed to avoid further non-zonal structures being induced besides to the enhanced GW forcing. The declination corresponds to January 15 (refers to the mid of the month) and the ozone and carbon dioxide data are taken from the year 2005 (refer to the mid of the decade). For the analysis, a time interval of 120 days with a temporal resolution of 2 hours after the spin-up period was modeled. Šácha et al. (2016) have

already analyzed the effect of the Asian hotspot with MUAM by performing a sensitivity study with regard to the strength of the GW forcing in the stratosphere. Their analysis time period was much shorter and the declination of the sun was different. They also nudged the model zonal mean temperature up to 30 km. Thus, the forcing due to the nudging slightly interacted with the implemented GW forcing, which is avoided in this new configuration here. We refer to this reference simulation as the 'Ref' simulation.

The state of the middle atmosphere of the Ref simulation can be seen in Fig. 1, which shows the January zonal mean zonal (a) and meridional wind (b), the temperature (c), zonal GW flux (d), the zonal wind acceleration due to breaking GWs (e), and the SPW 1 amplitude extracted from the zonal wind (f) as latitude-height plots. Each parameter is presented up to an altitude of 120 km for the winter and summer hemisphere. The zonal wind in Fig. 1(a) generally reproduces reference climatologies like CIRA-86 (Fleming et al., 1988) or URAP (Swinbank and Ortland, 2003), but the winter mesospheric jet is overestimated by

about 10-20 $ms^{-1}$. The meridional circulation (Fig. 1(b)) extending from the summer to the winter mesopause has a maximum of 6 $ms^{-1}$ at about 80 km, which is well reproducing predictions by climatologies (Portnyagin et al., 2004; Jacobi et al., 2009). Temperature (Fig. 1(c)) generally reproduce climatology values. The GW fluxes (Fig. 1(d)) maximize at about 80 km, with a maximum of slightly above -4 $m^2s^{-2}$ on the northern hemisphere (NH) and 2 $m^2s^{-2}$ on the southern hemisphere (SH). The corresponding zonal GW drag maximizes at the same altitude with about -60 $ms^{-1}day^{-1}$ (40 $ms^{-1}day^{-1}$) in the NH (SH) and

is westward (eastward) directed.

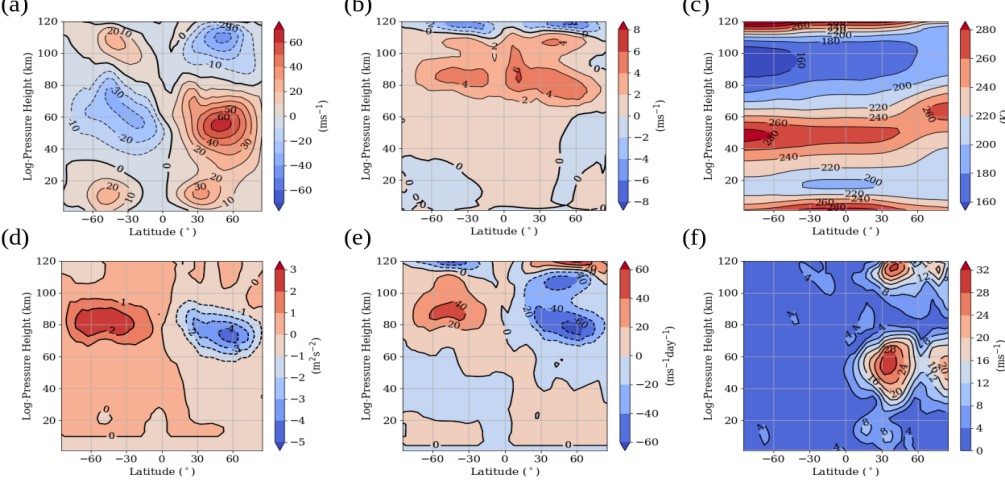

**Figure 1.** January zonal and monthly mean of the (a) zonal wind ($ms^{-1}$), (b) meridional wind ($ms^{-1}$), (c) temperature (K), (d) zonal GW fluxes ($m^2s^{-2}$), (e) zonal wind acceleration through breaking GWs ($ms^{-1}day^{-1}$) and (f) SPW 1 amplitude ($ms^{-1}$) extracted from the zonal wind of the reference simulation.

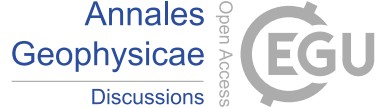



The SPW 1 amplitude (Fig. 1(f)) extracted from the zonal wind shows maximum values at the border of the mesospheric jet maximum northward of 30°N between 50 and 60 km and in the polar region. This fits quite well to observations but the amplitudes are slightly underestimated due to the overestimated mesospheric jet filtering some of the SPWs (Xiao et al., 2009).

## 2.2 Experiment description

In a first experiment, we reproduced the experiment of Šácha et al. (2016) to check if we still get similar results with the slightly modified setup. To represent the Asian GW breaking hotspot in the model we enhanced the GW drag after model day 270, i.e. after the spin up, and run the model for 120 days as in the Ref simulation. Therefore, the zonal ($GWD_u$) and meridional ($GWD_v$) GW drag and the heating due to breaking GWs ($GWD_T$) are modified in the specific region of the observed GW breaking hotspot. Like in Šácha et al. (2016) we located the GW breaking hotspot between 37.5°N-62.5°N and 118.1°E-

174.3°E in an altitude range between 18 and 30 km. Note that the geographic positions refer to the model grid points so that at a latitudinal 5° grid the meridional size of the modeled hotspot is 30°. To avoid a total breakdown of the polar vortex and a fundamental change in middle atmosphere dynamics, which was already forced in the study by Šácha et al. (2016), we chose the more moderate case of -10 ms$^{-1}$day$^{-1}$ for $GWD_u$, -0.1 ms$^{-1}$day$^{-1}$ for $GWD_v$ and a warming of 0.05 Kday$^{-1}$ for $GWD_T$. We refer to this simulation as the H3 simulation, as will be described later. The distribution of the $GWD_u$ of the Ref

and the H3 simulation can be seen in the left and middle panel of Fig. 2 at an altitude of about 27 km for the last 30 days of analysis. We are mainly concentrating on the last 30 days of analysis because we are focusing on quasi steady states and are not interested in short term variabilities. The $GWD_u$ of the Ref simulation is varying between -0.025 and +0.02 ms$^{-1}$day$^{-1}$ in the region of the EA/NP GW hotspot (37.5°N-62.5°N, 118.1°E-174.3°E, 18-30 km). Thus, the maximum value of the H3 simulation ($GWD_u$= -10 ms$^{-1}$day$^{-1}$ in the hotspot) is 500 times larger than the maximum westward (negative) value of the

Ref simulation. The H3 mean value (mean $GWD_u$: -10 ms$^{-1}$day$^{-1}$) is roughly 3300 larger than the one of the Ref simulation (mean $GWD_u$: 0.003 ms$^{-1}$day$^{-1}$) within the region of the EA/NP hotspot. In spite of the huge difference compared to the Ref simulation, the zonal GW forcing is moderate in terms of what is estimated from observations (40 ms$^{-1}$day$^{-1}$ and more) and from GW parameterizations in this region (Šácha et al., 2018). Concerning the meridional GW drag and the heating due to

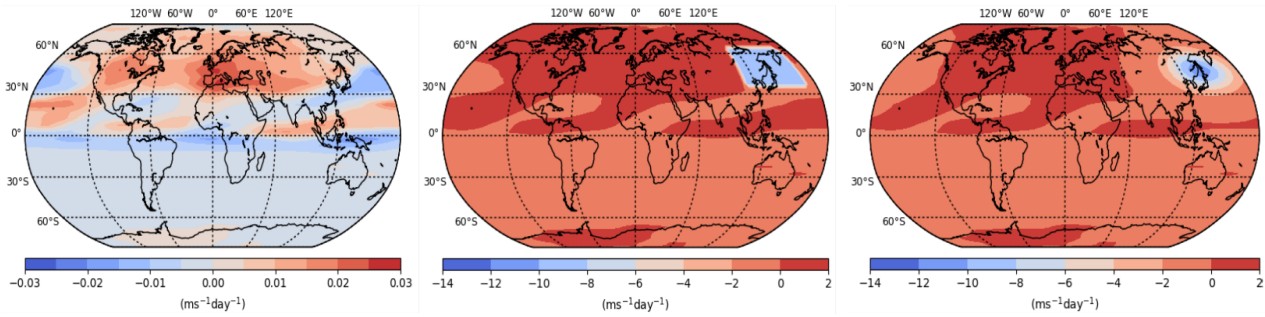

**Figure 2.** Zonal GW drag (ms$^{-1}$day$^{-1}$) at 26.9 km for the reference (left) and the H3 hotspot simulation as box (middle) and as Gaussian distribution (right) for the last 30 days of analysis. Note the different scaling on the left panel.





breaking GWs the maximum (mean) value of the H3 simulation is only 5 (100) times larger than the one of the Ref simulation (not shown here).

To investigate possible effects with regard to the position of the GW hotspot we performed a sensitivity study. For this, we kept the longitude (118.1°E-174.3°E) and altitude (18-30 km) range as well as the zonal extent of 25° fixed, but varied the

observed GW hotspot in 5° steps from 27.5°N-52.5°N (simulation H1) to 62.5°N-87.5°N (simulation H8), while labeling the experiments inbetween by H2 through H7. To analyze possible effects of the sharp transition zone between the unchanged and enhanced GW drag, additional simulations with a smoothed GW forcing were performed, by using a 3D Gaussian function, with a standard deviation of 10°, 22.5°, and 5.684 km in the zonal, meridional, and vertical direction, respectively. To get the same integral forcing as in the H1-H8 simulations, the maximum values for the $\text{GWD}_u$, $\text{GWD}_v$, and $\text{GWD}_T$ forcing have been

chosen as -13 ms$^{-1}$day$^{-1}$, -0.13 ms$^{-1}$day$^{-1}$, and 0.065 Kday$^{-1}$, respectively. For comparison, we are just focussing on the H3 hotspots with Gaussian smoothed boundaries in this paper.

## 3   Results

### 3.1   Hotspot effect on the background circulation

Fig. 3 shows the zonal mean zonal wind difference between each GW hotspot simulation H1-H8 (a-h) and the Ref simulation

in colour and the zonal mean zonal wind of the Ref simulation as contour lines in a latitude-height plot. The position of each GW hotspot is illustrated by the red box.

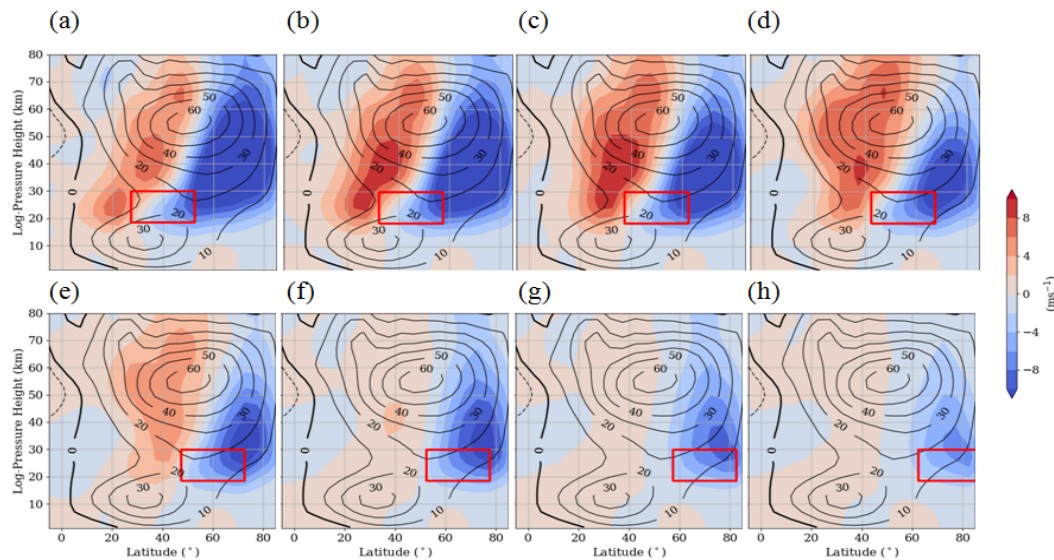

**Figure 3.** Zonal mean zonal wind difference between the H1-H8 (a-h) and the reference simulation (H1-8 - Ref). Color indicates the difference between the simulations and the contour lines show the zonal mean zonal wind of the reference simulation. Figures represent the last 30 days of the simulations. The position of each GW hotspot is represented by a red box.





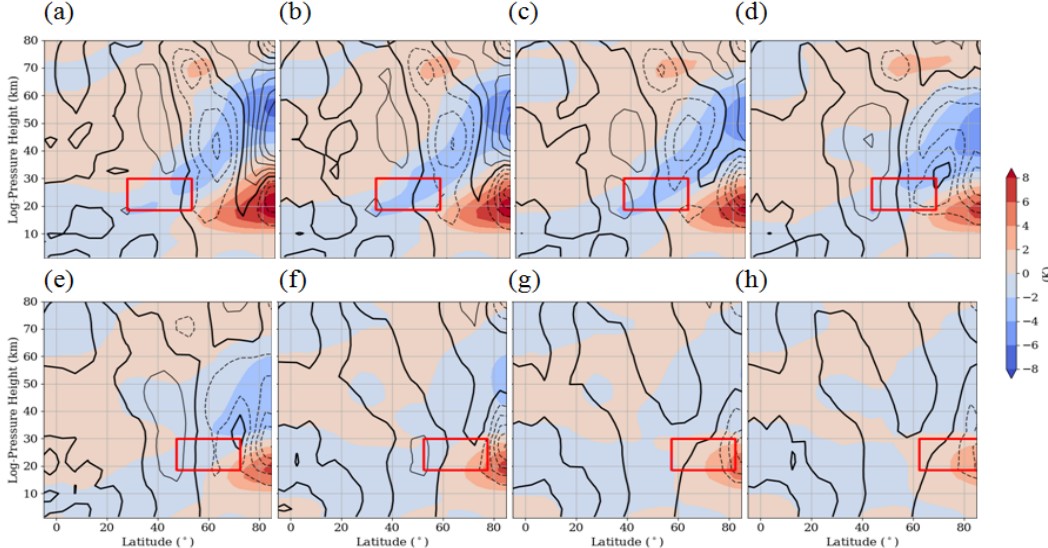

**Figure 4.** Zonal mean temperature and vertical wind difference between the H1-H8 (a-h) and the reference simulation (H1-8 – Ref). Color indicates the temperature difference between the simulations and the contour lines show the vertical wind difference starting with +/- 0.0005 ms$^{-1}$ up to +/- 0.0025 ms$^{-1}$ with increments of 0.0005 ms$^{-1}$ and a thicker zero line. Negative (positive) values are shown by dashed (solid lines). Figures represent the last 30 days of the simulations. The position of each GW hotspot is represented by a red box.

All experiments (H1-H8) show negative zonal wind differences with a maximum wind decrease of more than -10 ms$^{-1}$ in the polar region. Positive differences can be observed equatorward from an imaginary line connecting the subtropical and polar night jet centers, with a maximum difference of 8 to 10 ms$^{-1}$. These zonal wind anomalies are consistent with a polar vortex is shifted towards lower latitudes, and the wind reversal in the mesosphere is shifted upwards at lower latitudes. The strongest

decrease of the zonal mean zonal wind in the polar region wind can be observed in the H1 simulation (a) and the strongest increase of zonal mean zonal wind at lower latitudes can be observed in the H3 simulation (c), the latter is corresponding to the observed Asian GW hotspot. For GW hotspots with a southern edge north of 50°N the polar vortex is only slightly displaced towards lower latitudes. Thus, the effect of GW hotspots at higher latitudes is less strong.

Fig. 4 is arranged in the same manner as Fig. 3 but shows the temperature difference in colour and the vertical wind difference

as contour lines. As expected, the temperature effect scales with the (both zonal and vertical) wind differences, so that the H1-H3 simulations in Fig. 4(a-c) show the strongest temperature anomalies, and these are once more decreasing in magnitude for northward shifted GW hotspots. Between 60°N and 90°N, the GW hotspot leads to a temperature increase at altitudes up to 30-35 km, but to a decrease above. The zonal mean vertical wind difference shows generally negative anomalies between 15 km and 30 km at higher latitudes, which indicates a stronger downward movement connected with an adiabatic warming in the

lower part of the polar stratosphere. Above 35-40 km, we observe a positive vertical wind anomaly for the H1-H3 simulations, i.e. the downward movement is reduced and is leading to an adiabatic cooling anomaly. For most of the simulations, the negative anomaly in the lower part of the stratosphere is stronger than the positive anomaly above 40 km, which goes with the



distribution of the temperature anomalies. In case of the H4 and H5 (Fig. 4e, f) simulations the vertical wind anomalies do not fit to the temperature anomalies. We observe an increased downward movement in a region, where the temperature is weakly decreasing.

## 3.2 Influence on the polar vortex and anomalous SPWs

From previous publications it is already known that a warming (cooling) of the high-latitude stratosphere (mesosphere) and related changes in the dynamics are generally connected with PW activity. This leads us to the hypothesis that the main GWD enhancement effect is through SPW modulation, and this will be investigated in this subsection. In Fig. 5 we show for the H3 (left), H7 (middle) and Ref simulation (right panel) the geopotential height as contour lines and the zonal wind in colour coding as polar plot at 35 km, i.e., 5 km above the region of GW forcing. The panels represent the last 30 days of analysis. The

position of each GW hotspot is illustrated by the boxes (H1 (black) - H8 (violet)) in the right panel of Fig. 5. The polar vortex of the Ref simulation is stable (not displaced or splitted) and located near the North Pole (right panel of Fig. 5). Between 30 and 55°N the zonal wind of the Ref simulation is easterly in one part of the EA/NP region due to the Aleutian high (AH). This means that the GW forcing, which is normally acting against the westerly zonal mean zonal wind, is locally strengthening the zonal wind. Between 55°N and 90°N there is a strong west wind between East Asia and Alaska, thus, the GW forcing there is

acting locally against the zonal wind. Most of the GW hotspots (H1-H5) are located in the transition zone between easterlies and westerlies, whereby the south-eastern (north-western) part of each GW hotspot is located within the region of the easterlies (westerlies). The part of the GW forcing which is located in the easterlies (westerlies) is decreasing (increasing) for the more northward shifted GW hotspots (right panel of Fig. 5). In the H3 simulation, half of the GW hotspot is located in the easterlies and the other half in the westerlies.

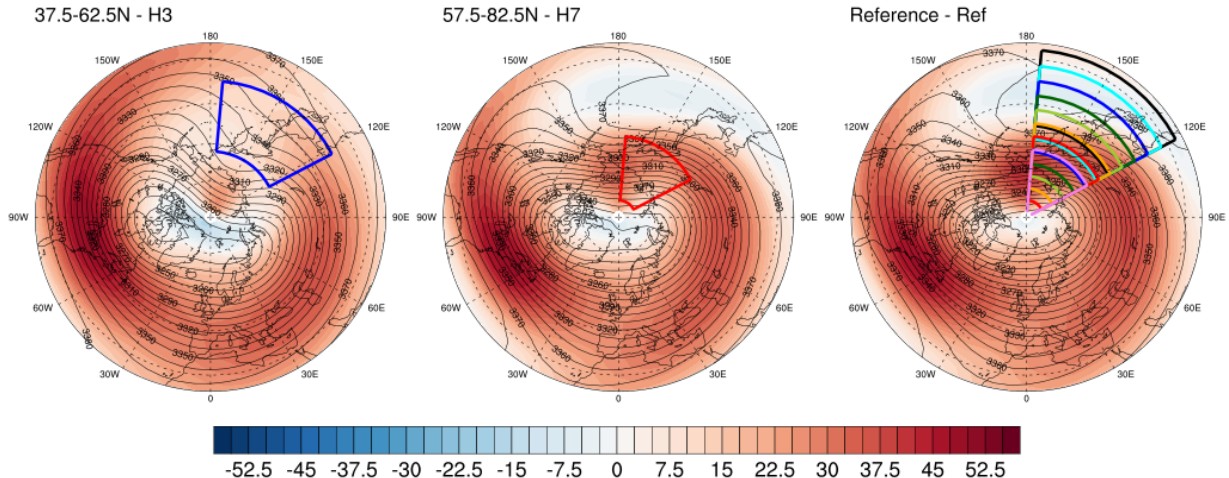

**Figure 5.** Zonal wind (ms$^{-1}$) in colour and geopotential height (gpdam) as contour lines northward of 25°N at 35 km for the H3 (left panel), H7 (middle panel) and the Ref simulation (right panel) representing the last 30 days of analysis. The boxes illustrate the position of each GW hotspot (H1 (in black) - H8 (in violet)). The blue (left and right panel) (red (middle and right panel)) box refers to the H3 (H7) simulation.

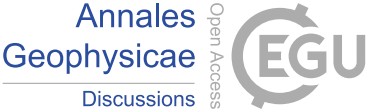


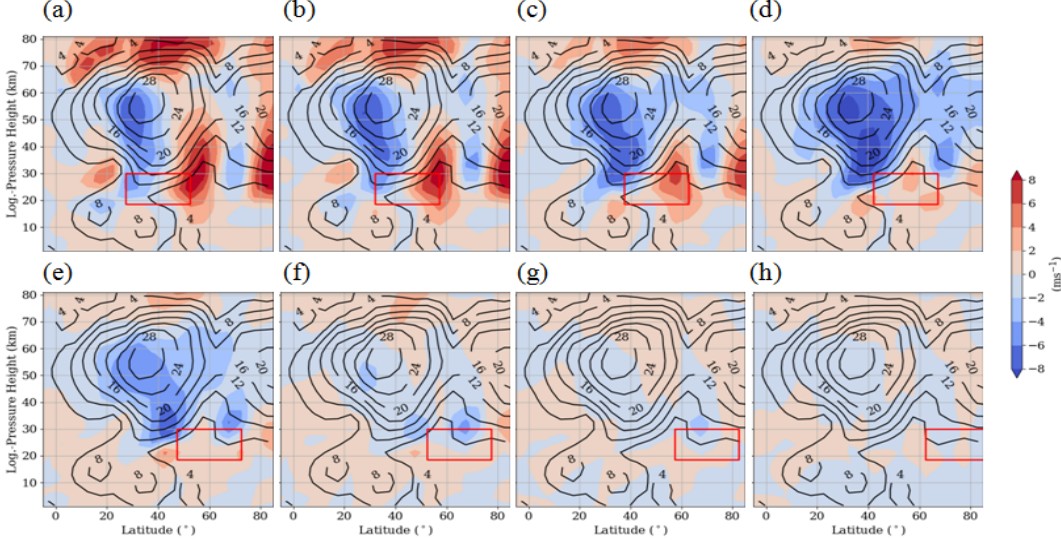

**Figure 6.** Zonal mean SPW 1 amplitude extracted from the zonal wind as difference between the H1-H8 (a-h) and the reference simulation. Color indicates the difference between the simulations and the contour lines show the zonal mean SPW 1 amplitude of the reference simulation. Figures represent the last 30 days of the simulations. The position of each GW hotspot is represented by the red box.

The state of the polar vortex of the H3 simulation is presented in the left panel of Fig. 5. The AH completely disappeared (no easterly wind anymore) and the center of the polar vortex is shifted towards Canada/Greenland. It is comma-like formed and slightly weaker and broader than the polar vortex of the Ref simulation. Thus, the H3 GW forcing has a destructive effect on the vortex circulation. This is in accordance with the results of the zonal wind differences (H3 - Ref) in Fig. 3(c) showing the

displacement of the polar vortex edge to lower latitudes. The H7 GW hotspot (middle panel of Fig. 5), which is completely located in the westerlies, the polar vortex is less disturbed by the GW forcing and remains nearly at the same position as in the Ref simulation. The non-zonal part of the zonal wind field which is mainly dominated by the SPW 1 will interact with the local zonal wind anomaly induced by the localized GW forcing. Since this zonal wind anomaly is localized in longitude, it may be decomposed into a spectrum of harmonics and can be assumed to be an additional wave interacting with the original

zonal wind SPW 1. To examine this interaction between the original SPW 1 in the model and the one induced by the local GW forcing the difference of the SPW 1 zonal wind amplitude between each GW hotspot simulation H1-H8 and the Ref simulation is shown in Fig. 6. The position of each GW hotspot is illustrated by the red boxes. In the Ref simulation the SPW 1 amplitude is maximizing at about 55 km between 30°N and 40°N with more than $28\,\mathrm{ms}^{-1}$ and also in the polar stratosphere with about $20\,\mathrm{ms}^{-1}$. In case of the H1-H4 simulations in Fig. 6(a-d) the zonal wind SPW 1 amplitude differences are positive (negative)

on the northern (southern) flank of the respective GW hotspot up to an altitude of about 60 km. The negative (positive) SPW 1 amplitude anomaly is increasing (decreasing) for more northward GW hotspots. The strongest increase (decrease) of SPW 1 amplitude can be observed in the H1 (H4) simulation with more than 8 (-8 $\mathrm{ms}^{-1}$). By comparing the positive and negative SPW 1 amplitude anomalies of the H1-H4 simulations it can be seen that the positive anomaly is less pronounced, while the



negative anomaly is more prevalent in the whole NH, particularly around the stratopause. The decreasing SPW 1 amplitude indicates that less SPWs 1 are propagating into the middle atmosphere. Due to the decreasing SPW 1 activity at lower latitudes, less SPWs 1 are breaking in this region, i.e. the zonal mean zonal wind is less decelerated as it is shown in Fig. 3.

Non-locally, however, a localized destructive (constructive) superposition of the original SPWs 1 within the model and the one

of the GW forcing may decrease (increase) the SPW 1 amplitude at other heights/latitudes due to changes of PW propagation. This effect can be observed around 55°N, where we observe an enhanced SPW 1 amplitude. It is strongest for the H1 GW hotspot and is decreasing for more northward located GW hotspots. The suppressed upward propagation of SPWs 1 is leading to the increasing SPW 1 amplitude in this area. This positive SPW 1 amplitude anomaly is corresponding to the decelerated zonal mean zonal wind in Fig.3. This leads to the assumption that the GW forcing may locally in- or decrease the SPW

1 amplitude but prevents the SPWs from propagating upwards into higher altitudes so that the SPW 1 amplitude is mainly decreasing in the stratosphere/mesosphere. Thus, the local GW forcing has a destructive effect on the circulation in the middle atmosphere. We will verify this in section 3.3 below by analysing the Eliassen-Palm flux.

Owing to the suppression of SPW 1 propagation at midlatitudes, the SPWs may increasingly propagate via the polar region, which may explain the increased SPW 1 amplitude in the polar stratosphere northward of 75°N. Another positive SPW 1

amplitude anomaly can be observed in the midlatitudinal mesosphere above 60 km which may be induced by local instabilities generating new SPWs 1. Both of these positive SPW 1 amplitude anomalies are strongest for the H1 simulation and are once more decreasing for northward shifted GW hotspots. The SPW 1 amplitude anomalies for the four northernmost GW hotspots H5-H8 in Fig. 6(e-h) are small in comparison to the four southernmost GW hotspot simulations, which corresponds to the observations in section 3.1. Only for the H5 simulation (Fig. 6(e)) the SPW 1 activity is also strongly reduced at lower latitudes

above 30 km like in the H1-H4 simulations.

To analyse in how far the GW forcing is locally affecting the SPWs of wavenumber 2 and 3, we compared the SPW 1 (left), 2 (middle), and 3 (right panel) amplitude anomalies at 35 km northward of 0°N/S (Fig. 7). The colours are the same as the colors of the hotspots in Fig. 5. As already discussed in Fig. 6, the SPW 1 amplitude locally increases at midlatitudes and in the polar region with about $10\,\mathrm{ms}^{-1}$ in maximum whereby the maxima are decreasing for northward displaced GW hotspots.

The negative anomaly, which is mainly dominating the middle atmosphere is located between 30°N and 40°N as well as at 70°N with more than $-10\,\mathrm{ms}^{-1}$ in minimum. The SPW 2 amplitude it can be seen that the SPW 2 activity is weakened/reduced northward of 30°N by about $-6\,\mathrm{ms}^{-1}$ in minimum. The largest decrease can be observed for the southernmost GW hotspot, and the decrease is getting smaller for northward displaced GW hotspots. Only at lower latitudes the SPW 2 amplitude is slightly increasing for those simulations. This is the case for the H2, H3, H4 and H5 simulation. The SPW 2 anomaly is negative

(positive) in those regions, where the SPW 1 anomaly is positive (negative). This leads to the assumption that just one of both SPWs (SPW 1 and SPW 2) can be the dominating one. By comparing the latitudinal distribution of the SPW 1 and 2 amplitude anomalies northward of 30°N it can be seen that they are similar when we neglect the scales. Both show a decrease in amplitude around 40°N and 70°N and an increase in the midlatitudes and in the polar region. In comparison, the SPW 3 amplitude anomaly distribution (Fig. 7 right) is slightly different because the SPW 3 amplitude is decreasing at 20°N (not at

40°N as for the SPW 1 and 2 anomalies). But, as for the SPW 1 and 2 anomalies, an increase of the SPW 3 amplitude induced



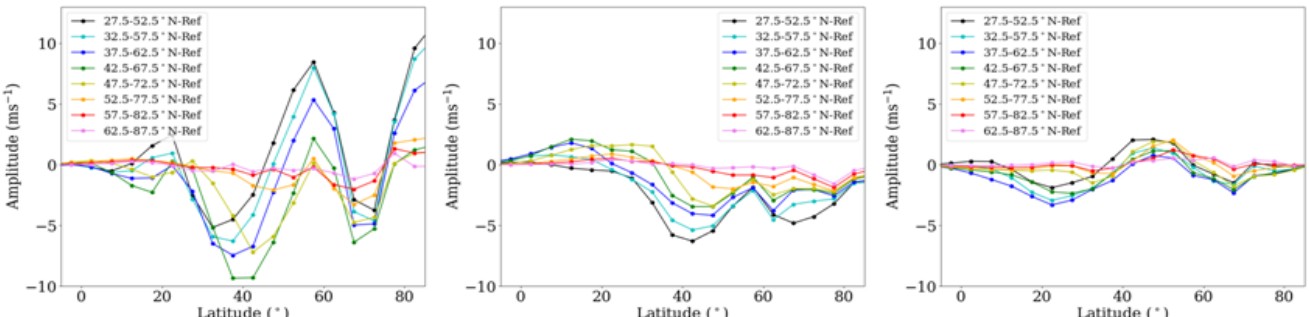

**Figure 7.** Zonal mean SPW 1 (left panel), SPW 2 (middle panel) and SPW 3 (right panel) amplitude as difference between the H1-H8 and the Ref simulation at 35 km for the last 30 days of the simulations extracted from the zonal wind.

by the local GW hotspots can be observed in the midlatitudes with about $2\,\mathrm{ms}^{-1}$ in maximum. The largest increase in SPW 3 amplitude can be seen in the H1 simulation (southernmost GW hotspot) and the largest decrease in the observed H3 simulation (observed Asian GW hotspot). The suppression of SPWs, which is induced by the local GW forcing, might be also an effect partly induced by the shape of the GW hotspot leading to a sharp transition zone between the unchanged and enhanced GW

5 drag values. To prove if the shape of the GW hotspot is partly leading to a suppression of SPWs, Fig. 8 shows the H3 amplitude anomaly from the left panel of Fig. 7 together with the corresponding Gauss simulation described in section 2. The latitudinal distribution of the SPW 1 amplitude difference is still the same showing the two local maxima at midlatitudes and in the polar region and the two minima at 40°N and 70°N but these two minima have changed from $-8\,\mathrm{ms}^{-1}$ for the three dimensional box to $-4\,\mathrm{ms}^{-1}$ for the Gaussian distribution. Also the maxima have changed from about $4\,\mathrm{ms}^{-1}$ to more than $5\,\mathrm{ms}^{-1}$. Due to

10 the stronger maximum GW drag in the Gaussian distribution more SPW 1 are excited, which is leading to the larger SPW 1 amplitudes at midlatitudes. The smoothly decreasing GW drag forcing towards lower and higher latitudes only slightly reduces the suppression of SPW 1 around 40°N and 70°N. Also, the mean wind and temperatures are only weakly affected, if we replace the box-like forcing by one with a Gaussian shape (not shown here). Thus, the GW hotspot itself is leading to essential changes in the dynamics suppressing the SPW propagation and decreasing the SPW 1 activity in the middle atmosphere.

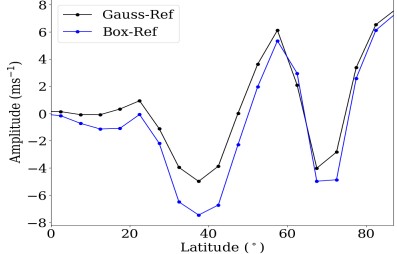

**Figure 8.** SPW 1 amplitude as difference between the H3 and the reference simulation (blue line) and the Gauss distribution and the reference simulation (black line) at 35 km for the last 30 days extracted from the zonal wind.



### 3.3 Propagation conditions for SPWs

To see in how far the SPW propagation is affected by the local GW forcing the EP fluxes and its divergence for SPW 1 were calculated. The results of the Ref simulation are presented in Fig. 9 (left panel). The arrows show the direction of propagation, the colour of the arrows represents the strength of the EP flux not normalized by the density, and the grey areas as well as the grey contour lines represent the EP divergence showing in which direction the zonal mean flow is accelerated. A negative (positive) EP divergence is illustrated by the dashed (solid) lines. The arrows were replaced by a dot when the amplitude is smaller than 1% of the maximum EP flux amplitude. The waves are mainly developing in the mid- and higher latitudes and from there they are propagating mainly towards the equatorial stratosphere/stratopause and, to a much lesser degree to the polar stratosphere. That the waves are really propagating upwards can be seen by means of the increasing amplitudes of the EP fluxes. The maximum EP flux amplitudes of more than $1.4\,\mathrm{m^2s^{-2}}$ are reached between 50°N and 60°N at an altitude of about 60 km, which corresponds to the height of the SPW 1 amplitude maximum in Fig. 1(f) of the Ref simulation.

In Fig. 10 the difference of the EP flux and its divergence between the H1-H8 and the Ref simulation is shown. The position of each GW hotspot is again illustrated by the red boxes. In the H1 and H2 simulations (Fig. 10(a-b)) more SPWs 1 are propagating into the polar stratosphere. These SPWs 1 are partly coming from the midlatitudes but most of them are directly generated in the polar region, where we observe a source of SPWs 1 (enhanced positive EP divergence at 70°N between 20 and 30 km). This positive EP divergence anomaly corresponds to the increased SPW 1 amplitude in the polar region in Fig. 7. Above this positive EP divergence anomaly an enhanced negative EP divergence is seen (from the northern flank of the GW hotspot up to 60 km tilted towards the North with increasing height), which means that the SPWs 1, which are propagating via the Arctic stratosphere, are breaking in this region. This leads to the deceleration of the zonal mean zonal wind in the middle and higher latitudes which was already discussed in Fig. 3. The negative EP divergence anomaly can also be seen in the H3-H5 simulations (Fig. 10(c-e)). This is the reason why the polar vortex is mainly disturbed by these GW hotspots (H1-H5).

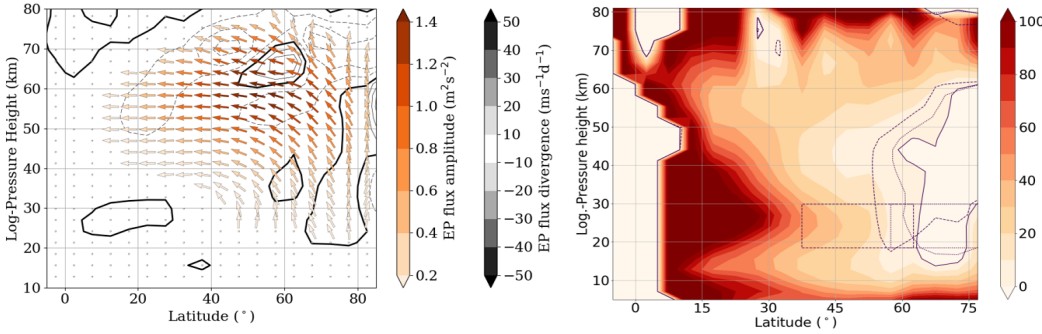

**Figure 9.** Zonal mean EP flux of SPW 1 of the Ref simulation (left panel). Contour lines show EP flux divergence, dashed lines denote negative EP flux divergence. Refractive index for SPW 1 for the Ref simulation (right panel) with a thicker zero line. The position of the H3 (H7) GW hotspot and the respective zero line is represented by the dashed (dotted) line colored in violet. Both Figs. represent the last 30 days of the simulation.




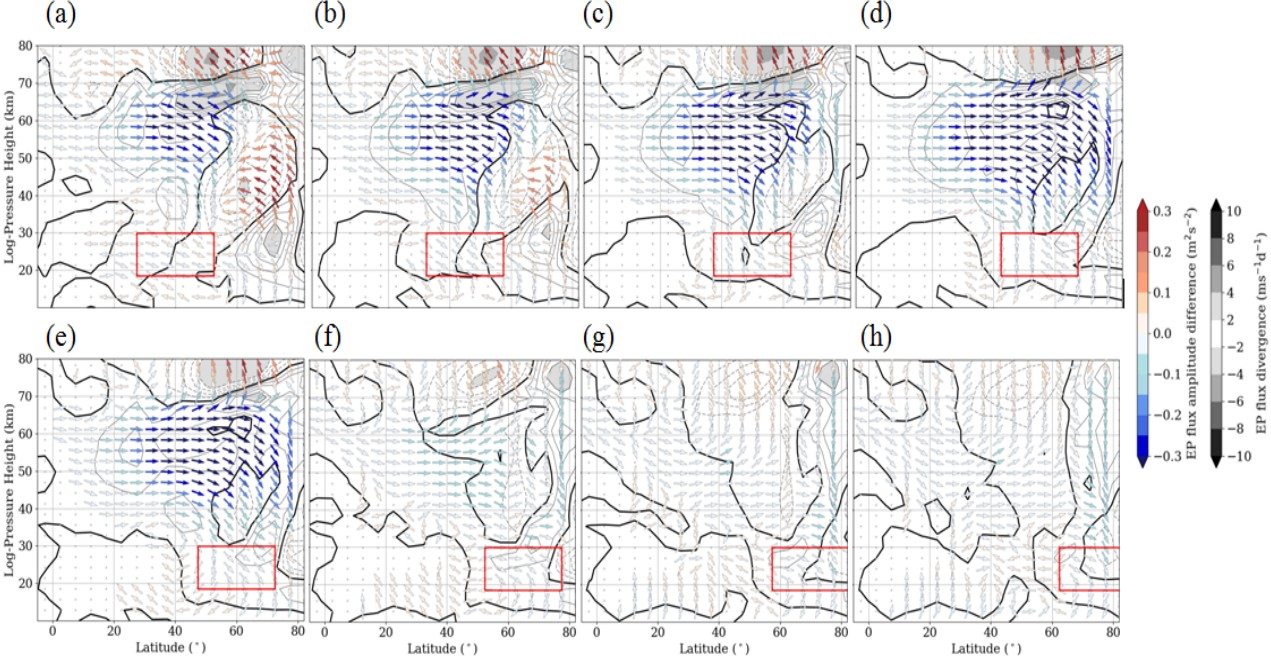

**Figure 10.** Zonal mean EP flux (arrows) and divergence (isolines and shaded areas, dashed lines show negative values) of SPW 1. Shown is the difference between all H1-8 (a-h) simulations and the reference simulation (H1-8 – Ref) representing the last 30 days of the simulations.

The negative EP divergence is strongest for the H1 simulation, which also exhibits the strongest increase of SPW 1 amplitude in the polar region. Furthermore, the H1-H5 simulations show a strong decrease of the EP flux amplitude (blue arrows) between 40 km and 70 km and 20°N to 80°N, which means that less SPWs 1 are propagating into the middle atmosphere. As a consequence, less SPWs 1 are breaking in this region leading to a positive EP divergence anomaly. This result corresponds

to the decreasing SPW 1 amplitude (Fig. 7) and the increasing zonal mean zonal wind (Fig. 3) at lower latitudes. The effect is strongest for the H4 simulation, which also shows the strongest decrease in SPW 1 amplitudes. Between 40°N and 70°N around 70 km we observe another source of SPWs 1, which propagate into the mesosphere, where these waves are breaking (strongly negative EP divergence above the positive EP divergence) due to the reversed wind conditions. The mesospheric EP flux in the H1-H5 simulations corresponds to the observed enhanced SPW 1 amplitude in the mesosphere in Fig. 7. Referring

to the enhanced SPW 1 around 55°N of the GW hotspots no enhanced EP flux can be observed in the respective region. But the arrows of the EP flux anomalies are pointing towards this area of enhanced SPW 1 amplitude. In the H6-H8 GW hotspot simulations (Fig. 10(f-h)) no large differences in EP flux and divergence occur, which corresponds to the small SPW 1 amplitude and the zonal mean zonal wind differences in Fig.7 and Fig.3. To explain why SPWs 1 do not propagate at higher latitudes also the refractive index (Matsuno, 1971; Andrews et al., 1987), multiplied by the square of the Earth´s radius $a^2$, is shown in Fig.

9 (right panel). The refractive index highly depends on the meridional potential vorticity gradient ($q_y$) and on the zonal mean zonal wind conditions (Li et al., 2007). White regions in Fig. 9 (right panel) indicate a negative refractive index, which means



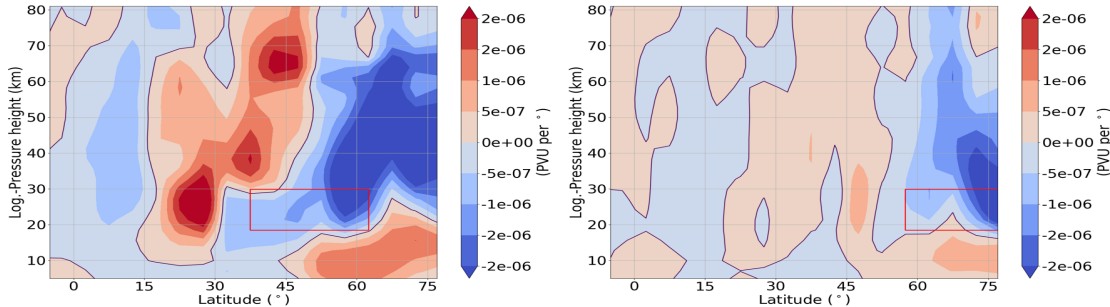

**Figure 11.** Meridional potential vorticity gradient difference between the H3 (left panel), H7 (right panel) and the reference simulation representing the last 30 days of the simulations. The positions of the H3 and H7 GW hotspot is represented by the red boxes.

that the waves cannot propagate in these regions. In the reddish regions wave propagation is possible. Due to the predominating west wind in the northern hemisphere the refractive index is mostly positive, so that SPWs 1 are able to propagate predominantly upward and towards the equator. Towards the midlatitudes and the polar region the refractive index is decreasing because of the increasing zonal mean zonal wind. The polar region (northward of 60°N) is the only region with a negative refractive

index in the northern hemisphere. This is because the polar vortex is a strong closed system, which repels most of the waves. To see in how far the refractive index is changing after implementing the GW forcing the position of the H3 (dashed line colored in violet) and the H7 (dotted line colored in violet) GW hotspot and the respective zero line of the refractive index were added to Fig. 9 (right panel). In the H3 simulation the zero line is higher in the polar region than the one in the Ref simulation. Thus, the refractive index is increasing (becomes more positive) in the polar region below 30 km, which corresponds to the enhanced

SPW 1 propagation and SPW 1 amplitude in the same region. The zero line of the H7 simulation is nearly at the same height as the one of the Ref simulation so that we do not observe huge changes in the Arctic. While the zero line of the Ref simulation is limited to the regions northward of 60°N, the zero line of the H3 (H7) simulation is located around 50°N (57°N). Based on the EP flux distribution of the Ref simulation in Fig. 9 (left panel) we already know that the SPWs 1 are mainly propagating from the midlatitudes (between 50 and 60°N) into the middle atmosphere. Due to the negative refractive index in this region

the SPWs 1 in the H3 and H7 simulations are not able anymore to propagate upwards so that the SPW 1 EP flux and amplitude are decreasing. Thus, the major branch of SPW 1 propagation is interrupted by the local GW forcing.

To check if there are local instabilities leading to the SPW 1 sources in the polar region and in the lower mesosphere, the $q_y$ differences between the H3 (H7) and the Ref simulation are shown in Fig. 11. The $q_y$ is given in potential vorticity units (PVU) per degree. The positions of the H3 and H7 GW hotspot are illustrated by the red boxes. Due to the increasing (decreasing)

zonal mean zonal wind at lower (higher) latitudes, the $q_y$, which is normally increasing towards higher latitudes, is reversed northward of 30°N. We observe a negative $q_y$ anomaly which is tilted towards the North with increasing height. Northward of 45°N up to 20 km the $q_y$ anomaly reverses again and becomes positive. These local reversals of the $q_y$, which are a necessary condition for baroclinic instability (Charney and Stern, 1962), can lead to the SPW 1 sources and positive EP divergences in the respective regions.





## 4   Conclusions

The sensitivity study regarding the effect of local GW hotspots in the stratosphere from lower to higher latitudes in a specific longitude range between 120°E and 170°E shows that GW hotspots southward of 50°N are leading to a negative refractive index at midlatitudes which prevents the SPWs from propagating upwards. Thus, less SPW 1 are breaking in the middle

atmosphere corresponding to the decreasing SPW 1 amplitude at lower latitudes connected with an increasing zonal mean zonal wind. Thus, the polar vortex is shifted towards lower latitudes but remains really strong (Baldwin and Holton, 1988) and leads additionaly to a suppression of SPWs according to the Charney-Drazin criterion (Charney and Drazin, 1961). The displacement of the polar vortex induced by breaking SPWs 1 causes an increase of the refractive index in the polar stratosphere (Karami et al., 2016) so that the SPWs 1 originating at midlatitudes are partly propagating via the polar region into the middle

atmosphere. Apart from these SPWs 1, additional SPWs 1 are propagating upwards which are directly generated in the Arctic owing to local baroclinic instability. One indication is the reversal of the $q_y$ (Charney and Stern, 1962; Garcia, 1991). For this reason we observe an enhanced EP flux and thus, an enhanced SPW 1 amplitude in the polar region. These SPWs 1 are breaking around 50 km between 50 and 80°N and lead to an enhanced negative EP divergence connected with a decreasing zonal mean zonal wind at higher latitudes. In the lower mesosphere between 40 and 70°N there is a second source of SPW 1

(positive EP divergence) developed as well due to local baroclinic instabilities (reversal of the $q_y$) (Smith, 2003; Lieberman et al., 2013; Matthias and Ern, 2018). As a consequence, the EP flux and the SPW 1 amplitude are enhanced between 70 and 80 km, right above the positive EP divergence anomaly. Based on the SPW 1 amplitude extracted from the zonal GW drag (not shown here) it was clear that each of the GW hotspots leads to a forcing of SPWs 1 but in some regions northward of 50°N this forcing is ineffective because the waves cannot propagate or are eliminated by destructive interference. The refractive index,

which is highly depending on the zonal mean zonal wind conditions, shows negative values in the polar region for the Ref simulation. So, if we implement a GW forcing directly in this region it has no impact on the middle atmosphere because SPWs cannot propagate. If we provoke a preconditioning of the polar vortex by first implementing e.g. the H1 GW hotspot and then adding one of the H6-H8 GW hotspots then the GW hotspots near the polar region would have a larger impact on the dynamics on the middle atmosphere.

Another interesting aspect is that different shapes of the local GW forcing will not have strong effects on the circulation. Despite of Gaussian-smoothed boundaries just negligible changes can be observed in the dynamics and SPW development, which are mainly due to the varying GW drag in the three dimensional Gaussian distribution leading to larger (smaller) effects when the Gaussian distribution is maximizing (minimizing).

Comparing the positions of these simulated GW hotspots with measurements (e.g. Hoffmann et al., 2013) it is clear that at

öeast some of the latitudianlly shifted GW hotspots are not very realistic so that our experiments should only be considered as a qualitative sensitivity study. To make it more realistic, the next step will be to analyse the effect of a longitudinally shifted hotspot (fixed latitude range between 30°N and 60°N), because observations and GCM experiments have shown their existence (listed in the introduction). In this latitude range GW hotpots like the Himalayan region, the Alps or the Rocky mountains are




part of the experiments. Also the interaction of two or more GW hotspots is part of our interests and will provide more insights into the effect of a localized GW forcing, which may be also important for the development of new GW parametrizations.

*Code availability.* MUAM model code is available from the corresponding author upon request.

*Author contributions.* N. Samtleben performed the MUAM model simulations and drafted the first version of the manuscript. P. Šácha
5    provided GW potential energy distributions. P. Pišoft, A. Kuchař and C. Jacobi actively contributed to the discussions and the writing of the paper.

*Competing interests.* C. Jacobi is one of the Editors-in-Chief of Annales Geophysicae. The authors declare that there are no conflicts of interest.

*Acknowledgements.* This study has been supported by Deutsche Forschungsgemeinschaft (DFG) under grant JA836/32-1, and by GA CR
10   under grant 16-01562J. ERA Interim reanalysis data have been provided by ECMWF through apps.ecmwf.int/datasets/data/. Petr Pišoft and Aleš Kuchař were supported by GA CR under grant nos. 16-01562J and 18-01625S. Petr Šácha was supported by GA CR under grant nos. 16-01562J and 18-01625S and by Government of Spain under grant no. CGL2015-71575-P and in later stages of the manuscript preparation through a postdoctoral grant of the Xunta de Galicia ED481B 2018/103. The EPhysLab is supported through the European Regional Development Fund (ERDF).



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



**Appendix A: Figures**

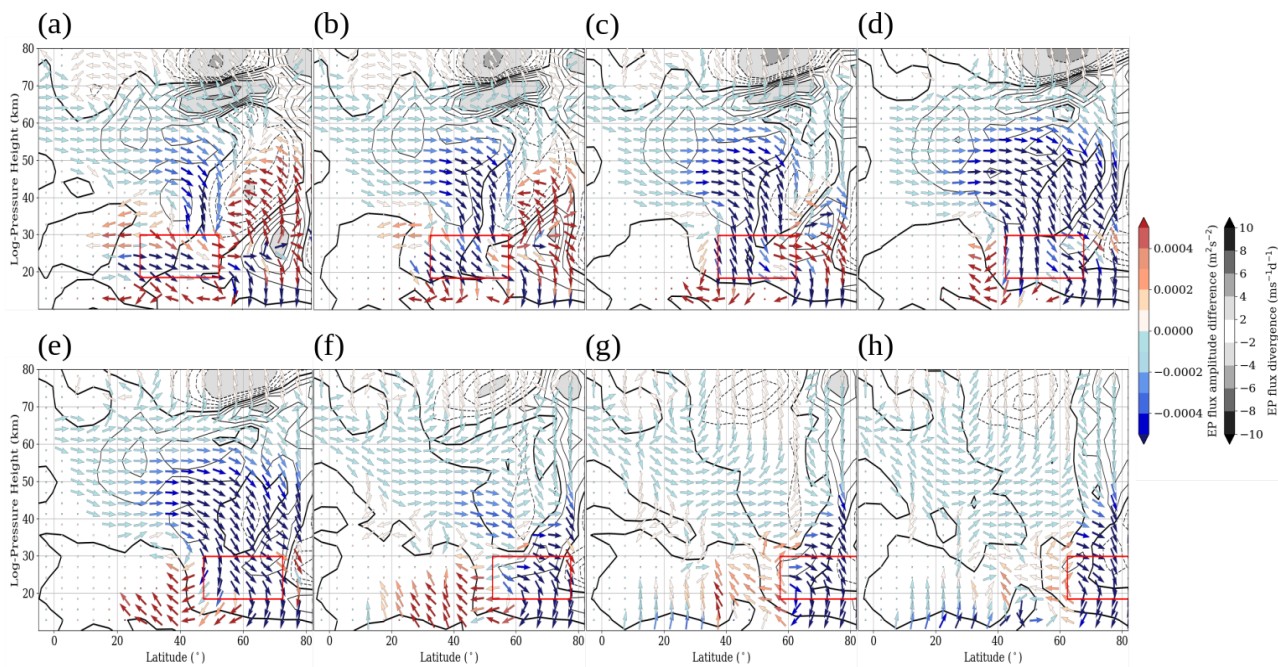

**Figure 12.** Zonal mean EP flux (arrows) and divergence (isolines and shaded areas, dashed lines show negative values) of SPW 1. Shown is the difference between all H1-8 (a-h) simulations and the reference simulation (H1-8 – Ref) representing the last 30 days of the simulations. The EP flux is weighted by the density.