# Peer review of "Effect of latitudinally displaced gravity wave forcing in the lower stratosphere on the polar vortex stability"

_Annales Geophysicae, 2019_

## Short Comment (SC1)

Dear reviewer #1,

we thank you for the comments and suggestions to improve the manuscript. Below we give some reply of the raised points, and will carefully consider all of them in the revised manuscript.

**In the paper the relevance of the work for the understanding of SSWs should be discussed (SSWs are not mentioned at all in the paper).**

Yes, we will do this in the revised version. Please also look at our response to comment no. 9.

**It should be qualitatively discussed how the model simulations agree with observations of gravity waves during SSWs.**

This is a good point, but it is hard to realize because we kept nearly all the conditions constant during our experiments. We save the GW drag data from the reference simulation, and then increased the GW drag in the specific region, and finally put it back into the model. We chose this cumbersome way, because otherwise we would get feedback mechanism which would in turn change the GW drag distribution. It was our intention to directly see the impact of this hotspot being not influenced by nonlinear effects. So, on the basis of the GW drag we do not see any changes. What is still changing is the horizontal GW momentum flux, which can be partly analyzed. It is just changing because of changes in the propagation conditions (background changes). The GW sources are fixed in MUAM thus, no additional GWs are generated. Ern et al. [2016] presented satellite measurements showing the absolute GW momentum flux during several years, also including SSWs. In general, they found out that the absolute GW momentum flux is increased (i) when the polar jet is strong and (ii) before and around the central day of a SSW and (iii) it is reduced when the zonal wind is weak. We tried to compare this to our GW output data containing the zonal and meridional GW flux, which we used to calculate the zonal mean absolute horizontal GW flux averaged between 60-80°N. It is scaled by density and presented in Fig. 1 in a time-height plot to show the temporal development. In contour lines the zonal mean zonal wind also averaged between 60 and 80°N is shown. The results in Fig. 1 are based on the H3 simulation (observed Asian GW hotspot). During the first 10 to 20 days the zonal mean zonal wind is decelerating, and the absolute horizontal GW flux decreases up to 70 km, which would correspond to the results from Ern et al. [2016]. However, in our simulation this effect is not very pronounced. We will discuss this in the revised version.

[Figure]

*Figure 1: Zonal mean absolute horizontal GW flux scaled by density (color coding) and the zonal mean zonal wind (contour lines) in a time-height plot during the analysed 120 model days. The data is based on the H3 simulation (Asian GW hotspot).*

**1. pg.2, line 2**
   **jet sources of gravity waves are not limited to frontal systems, and also the reference Plougonven et al. addresses jet sources of gravity waves more generally**

We agree that 'frontal systems' is too specific in this case. It is one source generating GWs owing to adjustment processes. Thus, it can be left out and we will include jet sources.

**2. pg.2, line 5**
**for the exponential increase with height constant background conditions have to be assumed and the gravity wave should propagate conservatively**

We will include these conditions in the text.

**3. pg.2, line 10**
**The expression "filtered out" could be misleading in this context. The waves moving into the direction of the wind, but faster, are not necessarily completely removed from the spectrum. Their amplitude growth, however, is hampered by their slow intrinsic phase speed.**

This is an infelicity of the expression at this point. When they are faster than the background then the GWs are of course not filtered out but if they reach the critical line (background flow equal to the phase speed – for most of the GWs at the latest in the altitude of the mesospheric jet) the GWs are absorbed by the background flow or reflected back to the troposphere. We will correct it by adding that these GWs are partly filtered out when the phase speed of the GWs is equal to the background flow (mainly in the region of the mesospheric jet).

**4. pg.2, line 10, suggestion:**
**"For this reason the wind reverses in the mesosphere due to GW breaking, while in the opposite direction to u travelling GWs deposit their momentum (Lindzen, 1981; Holton, 1982)."**
**"In mesosphere even GWs propagating in the opposite direction to u saturate and deposit their momentum. For this reason the wind reverses in the upper mesosphere/lower thermosphere (Lindzen, 1981; Holton, 1982)."**

Thank you for this suggestion. Our sentence contains a huge amount of information and can be quite confusing wherefore your version is much better and we will include this one in our paper.

**5. pg.2, line 20/21**
**Compared to the enumeration of mountain wave sources, the enumeration of convective regions generating gravity waves is quite short. Suggestion for keeping the balance:**
**"Typically, in the stratosphere satellite observations show a characteristic structure of enhanced GW activity in the subtropics that is caused by deep convection over Southeast Asia, America, Africa, or the Maritime Continent in the respective summer season (Jiang et al., 2004; Ern and Preusse, 2012; Wright and Gille, 2011)."**

Thank you for this suggestions. This is a good idea to keep them in balance. Also because our observed Asian GW hotspot is partly convectively and orographically generated so that we should represent both trigger mechanisms (convection and orography) equally.

**6. pg.2, line 22**
**Here you write "...objective determination of the GW drag from satellite measurements alone is not possible."**
**This is a too strong statement that also does not hold. Estimates based on satellite data exist, however contain large errors.**
**In the text following line 22 you give two references and state that GW drag would be derived from superpressure balloons and lidar data. However, these two references are used out-of-context!**
**The GW drag in these papers is based on model simulations alone. So far GW drag has not been derived from superpressure balloons because these balloons float on a fixed altitude, and vertical gradients cannot be inferred. To my knowledge, GW drag has not been tried from lidar data (please correct me if I am wrong), but I know that GW drag has been derived from radar data. Similar as GW drag derived from satellite, these estimates generally have large errors.**

**Therefore I would suggest the following rewording:**
**"Reliable estimates of GW drag from observations are generally difficult. There are attempts from satellite (for example, Ern et al., 2014; Ern et al., 2016), or from radar (for example, Reid and Vincent, 1987). Usually, however, uncertainties of these estimates are quite large. From model studies there are indications that GWs can break already in the lower stratosphere (LS), for example Plougonven et al. (2008), or Constantino et al. (2015)."**

We agree with your point of view. Maybe our statement is a bit misleading because we meant the measurement of the GW drag itself. As you said, it is possible to derive the drag by making different assumptions which create large errors. We will change this part based on your suggested text.

**7. pg.3, line 29**
**This comment is not relevant for the current paper, but may become relevant for future work. On pg.3 line 29 you write that GW potential energy of the MUAM model would be tuned "... based on potential energy data obtained from GPS radio occultation measurements" Tuning the model towards GW potential energies observed from GPS may introduce large biases. It has been shown by Rapp et al. (2018) that vertical filtering of GPS soundings for obtaining the gravity wave signal does not remove larger scale structures having short vertical wavelengths, such as inertial instabilities at low and mid latitudes, or Kelvin waves in the tropics (Ern et al., 2008).**

Yes, that's true. This is something we kept in mind when we were analyzing the GPS RO data which as well do not consider Kelvin waves, for example. Before we included the globally weighted potential energy data as zonal mean into our model, we had an artificial zonal mean GW amplitude distribution based on a hyperbolic tangent function depending on the latitude which was less realistic then the one derived from the GPS RO data. We thought it is a good idea to replace it but of course we should not forget that these data does not include all small-scale disturbances. We will discuss this further in the revised version.

**8. pg.15, line 30**
**Gravity wave hotspots do not only occur over mountain ridges. Local hotspots can also be caused by the GW jet source mechanism. For example, Ern et al., 2016 find enhancements of GW activity that are linked to jet exit regions. Possibly these hotspots are missed by AIRS because of its limitation to gravity waves of very long vertical wavelengths. The proposed follow-up study is therefore a promising way to proceed, but should not be limited to the positions of orographic GW hotspots. Also longitudinally varying positions of jet-related GW sources could be important.**

Thank you for this suggestion. In this paper the position of the analyzed GW hotspots does not explicitly refer to a specific source. Regarding orography there are no obvious sources, when we are displacing the GW hotspot latitudinally. But of course, some of these GW hotspots can be purely hypothetical connected to jet exit regions.
For this purpose, we are planning to write another paper about the longitudinal displacement of the GW hotspot. When we are displacing the GW hotspot longitudinally then we do not only capture/represent orographically generated GWs. Because this displacement is partly along the polar front jet which may generate GWs we indirectly consider this kind of GWs. Then, we will also include this type of GW as possible contributor of the artificial GW forcing.

**9. General comment for the discussion throughout the paper:**
**For midlatitude forcing some of the findings are very similar to the situation during SSWs.**
**- vortex slowdown and shift (Fig. 3)**
**- stratospheric warming (Fig. 4)**
**- changes in the activity of the SPW1**
**These similarities should be discussed in more detail and put into the context of SSWs. For example, it has been suggested recently by Albers and Birner (2014) that gravity wave forcing at**

**midlatitudes could be important for the onset of SSWs. Also satellite observations of gravity waves show stronger GW drag at mid or even low latitudes prior to major SSW central dates (Ern et al., 2016). In the same paper strong gravity wave activity and GW drag was seen in the polar vortex during 2011. In this year, however, the vortex was very stable and confined to high latitudes which also confined gravity wave activity to high latitudes, apparently without affecting the stability of the vortex. This supports one of the main findings of your study and should therefore be mentioned.**

Thank you for this comment! Indeed, the results show specific characteristics which are comparable to SSWs. In this case, we did not create a SSW but caused a preconditioning of the polar vortex owing to the aspects (SPW activity, displacement and slight warming) you already mentioned. But this also strongly depends on the strength of the forcing (stronger forcing can also lead to a total breakdown of the polar vortex). Based on this fact we will include this topic (preconditioning of the polar vortex) in our discussion and will also refer to this topic in the introduction.

**Other Comments**
**1. pg.2, lline 28/29: Please rewrite the following sentence for better legibility!**
   **"This study is focusing on different GW breaking areas in the lower stratosphere along and the effects on the middle atmosphere highly depending on the position."**

We will rewrite this part such as: this study is focusing on different GW breaking areas in the lower stratosphere and their effect on the middle atmosphere dynamics.

**2. pg.2, line 29: who were concentrating ! who were focusing**

We will replace the word as suggested.

**3. pg.3, line 2: shifting meridionally the EA/NP hotspot along its fixed longitude range! shifting meridionally the EA/NP hotspot keeping its longitude range fixed**

We will change this sentence as suggested.

**4. pg.3, line 23: interval, in which ! interval in which**

We will correct this.

**5. pg.4, line 3: refer to the mid ! referring to the mid**

We will correct this.

**6. pg.4, line 17: reproduce ! reproduces**

We will correct this.

**7. pg.5, line 2: observations but the ! observations, but the**

We will correct this.

**8. pg.5, line 3: jet filtering some of ! jet filtering of some of**

We will modify this sentence.

**9. pg.7, line 4: is shifted toward ! that is shifted toward**

We will correct this.

**10. pg.7, line 14: kmat ! km at**

We will correct this.

**11. pg.8, line 14: west wind ! westerly wind**

We will correct this.

**12. pg.11, line 10: more SPW 1 are excited, ! SPW 1 excitation is strengthened,**

We will correct this.

**13. pg.12, line 2: fluxes and its ! fluxes and their**

We will correct this.

**14. pg.13, line 3: atmopshere. ! atmosphere.**

We will correct this.

---

## Short Comment (SC2)

Dear reviewer #2,

we thank you for the comments and suggestions to improve the manuscript. Below we give some reply of the raised points, and will carefully consider all of them in the revised manuscript.

**The authors should include a table summarizing the different experiments.**

We agree with the referee. Owing to the huge amount of different experiments it would be better to summarize them in a table. This table may include the position as well as the different GW forcings in the respective region, e.g. as in Table 1 below.

| Simulation | Abbreviation | Region                                  | Max. zonal
GW drag
(ms -1 day -1 ) | Mean zonal
GW drag
(ms -1 day -1 ) | Max.
meridional
GW drag
(ms -1 day -1 ) | Mean
merdidional
GW drag
(ms -1 day -1 ) | Max. heating
by GWs
(Kday -1 ) | Mean
heating
by GWs
(Kdav -1 ) |
|------------|--------------|-----------------------------------------|----------------------------------------------------------------|----------------------------------------------------------------|------------------------------------------------------------------------|-------------------------------------------------------------------------|-------------------------------------------------|----------------------------------------------------|
|            |              |                                         |                                                                |                                                                | (                                                                      | (1110 111)))                                                            |                                                 | (====;;) )                                         |
| Reference  | Ref          |                                         | -0.025/0.02                                                    | 0.003                                                          | -0.025/0.01                                                            | -0.001                                                                  | -0.0006/0.01                                    | 0.0003                                             |
| Hotspot    | H1-H8        | 27.5-87.5°N
118.1-174.3°E
18-30km | -10                                                            | -10                                                            | -0.1                                                                   | -0.1                                                                    | 0.05                                            | 0.05                                               |
| Gauss      | Gauss        |                                         | -13.1                                                          |                                                                | -0.13                                                                  |                                                                         | 0.065                                           |                                                    |

Table 1: Overview of the mean and maximum values of the zonal and meridional GW drag and heating by GWs for the reference and hotspot simulations as three dimensional box and as Gaussian distribution. The mean and maximum values refer to the region (118.1-174.3°E, 18-30km and the respective latitude range) of the hotspot.

The reviewer found it a bit confusing that the Gaussian GW hotspot is shown in Figure 2 and the comparison between the 'used' simulations and the Gaussian is later presented in Figure 8. It is suggested to present this type of sensitivity in the experimental description section and later just refer to the 'main' model experiments.

We agree that it might be a bit confusing. However, to evaluate the possible effect of a Gaussian distribution we first have to analyze at least the differences of the three-dimensional box H3 GW hotspot and the Ref simulation. We will add more explanations to the description of Figure 2 and the new Table 1, and clearly state that we will come back to the Gaussian-shaped hotspot later.

**Please clarify whether the altitudes shown in the Figures correspond to the pressure grid or whether the altitudes are computed from the geopotential and converted into a geometric altitude. This simplifies some comparisons to observations results.**

The altitude, which is shown or chosen in all figures is given in logarithmic pressure height. It is defined by:

$$z = -Hln\left(\frac{p}{p_0}\right)$$

| Ζ                         | logarithmic pressure height |
|---------------------------|-----------------------------|
| H = 7 km                  | scale height                |
| p                         | pressure at specific level  |
| p 0 = 1000 hPa | reference pressure          |

The logarithmic pressure height corresponds to the geometrical height up to an altitude of about 110 km with small deviations increasing within the altitude range. In 110 km the deviation is about 5 km. Depending on the thermospheric temperature the logarithmic pressure height can strongly differ from the geometrical height. Because we are just interested in altitudes up to 80 km, it can be assumed that the given altitude is close to the geometric height. We will add an explanation in section 2: "Note that depending on the temperature profile the used logarithmic pressure height can differ from the geometric heights. However, at altitudes below 80 km, this difference is negligibly small."

Another comment of the reviewer concerns the implementation of the GW drag for the different latitudes. As the drag scales also with the atmospheric air mass that is affected by the drag, it might be mentioned that at higher latitudes essentially less drag is exerted to the atmosphere as the GW drag volume scales with latitude. Or with other words the atmospheric mass that is affected by the drag decreases with latitude. This might need some more discussion or should at least be mentioned in the interpretation of the results.

The reviewer is right, the integrated drag scales with the cosine of latitude. This is, however, necessary, because otherwise the drag, both locally and as a zonal mean, would strongly increase with latitude. But in the governing equations such as e.g. the continuity equation the horizontal winds are scaled as well, so that this would result in a different experiment. We will add a note on this in section 2.

**Some Figures (3,4,6 and 7) need an improvement of the quality.**

We will provide them in higher resolution.

**Page 3: line 18: . . .vertical resolution 2842 km should be 2842 m**

We will correct that.

---

## Referee Comment (RC1) · Anonymous Referee #1 · 28 Feb 2019

This paper is a sensitivity study about the effect of locally enhanced forcing by gravity waves on the polar vortex. In the mechanistic global circulation model MUAM a longitudinally and vertically confined forcing region is shifted towards lower and higher latitudes in steps of 5deg. The main results are that (1) forcing at higher latitudes has little effect on the stability of the vortex, and (2) forcing at lower latitudes leads to breaking of planetary waves and a slowdown and displacement of the vortex.

This study is very interesting because it contributes to an active field of research that addresses the question how sudden stratospheric warmings (SSWs) are triggered. This topic is of interest to the broad community of atmospheric and climate research

because SSWs are known to impact weather and climate at the Earth's surface.

Publication of the paper in Annales Geophysicae is therefore highly recommended.

There is, however, one major issue that should be addressed before publication:
(1a) in the paper the relevance of the work for the understanding of SSWs should be discussed (SSWs are not mentioned at all in the paper)
(1b) it should be qualitatively discussed how the model simulations agree with observations of gravity waves during SSWs

Addressing this issue should however be possible with reasonable effort.

A list of specific and other comments is given in the following.

**Specific Comments**

1. pg.2, line 2
   jet sources of gravity waves are not limited to frontal systems, and also the reference Plougonven et al. addresses jet sources of gravity waves more generally

2. pg.2, line 5
   for the exponential increase with height constant background conditions have to be assumed and the gravity wave should propagate conservatively

3. pg.2, line 10
   The expression "filtered out" could be misleading in this context. The waves moving into the direction of the wind, but faster, are not necessarily completely removed from the spectrum. Their amplitude growth, however, is hampered by their slow intrinsic phase speed.

4. pg.2, line 10, suggestion:

"For this reason the wind reverses in the mesosphere due to GW breaking, while in the opposite direction to u travelling GWs deposit their momentum (Lindzen, 1981; Holton, 1982)."

→

"In mesosphere even GWs propagating in the opposite direction to u saturate and deposit their momentum. For this reason the wind reverses in the upper mesosphere/lower thermosphere (Lindzen, 1981; Holton, 1982)."

5. pg.2, line 20/21

Compared to the enumeration of mountain wave sources, the enumeration of convective regions generating gravity waves is quite short. Suggestion for keeping the balance:

"Typically, in the stratosphere satellite observations show a characteristic structure of enhanced GW activity in the subtropics that is caused by deep convection over Southeast Asia, America, Africa, or the Maritime Continent in the respective summer season (Jiang et al., 2004; Ern and Preusse, 2012; Wright and Gille, 2011)."

References:

Jiang, J. H., B. Wang, K. Goya, K. Hocke, S. D. Eckermann, J. Ma, D. L. Wu, and W. J. Read (2004), Geographical distribution and interseasonal variability of tropical deep convection: UARS MLS observations and analyses, J. Geophys. Res., 109, D03111, doi:10.1029/2003JD003756.

Ern, M., and P. Preusse (2012), Gravity wave momentum flux spectra observed from satellite in the summertime subtropics: Implications for global modeling, Geophys. Res. Lett., 39, L15810, doi:10.1029/2012GL052659.

Wright, C. J., and J. C. Gille (2011), HIRDLS observations of gravity wave momentum fluxes over the monsoon regions, J. Geophys. Res., 116, D12103, doi:10.1029/2011JD015725.

6. pg.2, line 22
   Here you write "...objective determination of the GW drag from satellite measurements alone is not possible."

   This is a too strong statement that also does not hold. Estimates based on satellite data exist, however contain large errors.
   In the text following line 22 you give two references and state that GW drag would be derived from superpressure balloons and lidar data.
   However, these two references are used out-of-context!
   The GW drag in these papers is based on model simulations alone. So far GW drag has not been derived from superpressure balloons because these balloons float on a fixed altitude, and vertical gradients cannot be inferred. To my knowledge, GW drag has not been tried from lidar data (please correct me if I am wrong), but I know that GW drag has been derived from radar data. Similar as GW drag derived from satellite, these estimates generally have large errors.

   Therefore I would suggest the following rewording:

   "Reliable estimates of GW drag from observations are generally difficult. There are attempts from satellite (for example, Ern et al., 2014; Ern et al., 2016), or from radar (for example, Reid and Vincent, 1987). Usually, however, uncertainties of these estimates are quite large. From model studies there are indications that GWs can break already in the lower stratosphere (LS), for example Plougonven et al. (2008), or Constantino et al. (2015)."

   References:

   Reid, I. M., and R. A. Vincent (1987), Measurements of mesospheric gravity wave momentum fluxes and mean flow accelerations at Adelaide, Australia, J. Atmos. Terr. Phys., 49, 443-460.

   Ern, M., F. Ploeger, P. Preusse, J. C. Gille, L. J. Gray, S. Kalisch, M. G. Mlynczak, J. M. Russell III, and M. Riese (2014), Interaction of gravity waves

with the QBO: A satellite perspective, J. Geophys. Res. Atmos., 119, 2329-2355, doi:10.1002/2013JD020731.

Ern, M., Trinh, Q. T., Kaufmann, M., Krisch, I., Preusse, P., Ungermann, J., Zhu, Y., Gille, J. C., Mlynczak, M. G., Russell III, J. M., Schwartz, M. J., and Riese, M.: Satellite observations of middle atmosphere gravity wave absolute momentum flux and of its vertical gradient during recent stratospheric warmings, Atmos. Chem. Phys., 16, 9983-10019, https://doi.org/10.5194/acp-16-9983-2016, 2016.

7. pg.3, line 29 — This comment is not relevant for the current paper, but may become relevant for future work.

On pg.3 line 29 you write that GW potential energy of the MUAM model would be tuned "... based on potential energy data obtained from GPS radio occultation measurements"

Tuning the model towards GW potential energies observed from GPS may introduce large biases. It has been shown by Rapp et al. (2018) that vertical filtering of GPS soundings for obtaining the gravity wave signal does not remove larger scale structures having short vertical wavelengths, such as inertial instabilities at low and mid latitudes, or Kelvin waves in the tropics (Ern et al., 2008).

References: Rapp, M., Dornbrack, A., & Preusse, P. (2018). Large midlatitude stratospheric temperature variability caused by inertial instability: A potential source of bias for gravity wave climatologies. Geophysical Research Letters, 45, 10682-10690. https://doi.org/10.1029/2018GL079142

Ern, M., Preusse, P., Krebsbach, M., Mlynczak, M. G., and Russell III, J. M.: Equatorial wave analysis from SABER and ECMWF temperatures, Atmos. Chem. Phys., 8, 845-869, https://doi.org/10.5194/acp-8-845-2008, 2008.

8. pg.15, line 30
Gravity wave hotspots do not only occur over mountain ridges. Local hotspots

can also be caused by the GW jet source mechanism. For example, Ern et al., 2016 find enhancements of GW activity that are linked to jet exit regions. Possibly these hotspots are missed by AIRS because of its limitation to gravity waves of very long vertical wavelengths. The proposed follow-up study is therefore a promising way to proceed, but should not be limited to the positions of orographic GW hotspots. Also longitudinally varying positions of jet-related GW sources could be important.

9. General comment for the discussion throughout the paper:
For midlatitude forcing some of the findings are very similar to the situation during SSWs.
- vortex slowdown and shift (Fig. 3)
- stratospheric warming (Fig. 4)
- changes in the activity of the SPW1

These similarities should be discussed in more detail and put into the context of SSWs. For example, it has been suggested recently by Albers and Birner (2014) that gravity wave forcing at midlatitudes could be important for the onset of SSWs. Also satellite observations of gravity waves show stronger GW drag at mid or even low latitudes prior to major SSW central dates (Ern et al., 2016). In the same paper strong gravity wave activity and GW drag was seen in the polar vortex during 2011. In this year, however, the vortex was very stable and confined to high latitudes which also confined gravity wave activity to high latitudes, apparently without affecting the stability of the vortex. This supports one of the main findings of your study and should therefore be mentioned.

Reference:

Albers, J. R., & Birner, T. (2014). Vortex preconditioning due to planetary and gravity waves prior to sudden stratospheric warmings, J. Atmos. Sci., 71, 4028-4054, doi:10.1175/JAS-D-14-0026.1.

**Other Comments**

1. pg.2, lline 28/29: Please rewrite the following sentence for better legibility!
   "This study is focusing on different GW breaking areas in the lower stratosphere
   along and the effects on the middle atmosphere highly depending on the position."

2. pg.2, line 29: who were concentrating → who were focusing

3. pg.3, line 2
   shifting meridionally the EA/NP hotspot along its fixed longitude range
   →
   shifting meridionally the EA/NP hotspot keeping its longitude range fixed

4. pg.3, line 23: interval, in which → interval in which

5. pg.4, line 3: refer to the mid → referring to the mid

6. pg.4, line 17: reproduce → reproduces

7. pg.5, line 2: observations but the → observations, but the

8. pg.5, line 3: jet filtering some of → jet filtering of some of

9. pg.7, line 4: is shifted toward → that is shifted toward

10. pg.7, line 14: kmat → km at

11. pg.8, line 14: west wind → westerly wind

12. pg.11, line 10: more SPW 1 are excited, → SPW 1 excitation is strengthened,

13. pg.12, line 2: fluxes and its → fluxes and their

14. pg.13, line 3: atmopshere. → atmosphere.

15. pg.14, line 2: west wind → westerly wind

16. pg.15, line 7: additionaly → additionally

17. pg.15, line 30: öeast → least

18. pg.15, line 30: latitudianlly → latitudinally

---

## Referee Comment (RC2) · Anonymous Referee #2 · 1 Apr 2019

General Comment: The authors present idealistic simulations with the mechanistic MUAM model testing the impact of stratospheric GW hotspot regions on the SPW generation and vertical propagation. Thus, they conducted several model runs modifying the gravity wave hotspot in the Asian sector (above Japan) location and investigate how it affected the polar vortex and the lower and mid-latitude dynamics. The results of this sensitivity study are discussed presenting various figures and diagnostics. The study is relevant to the community as this case study provides a potential connection how gravity waves excited in the stratospheric above a certain region can lead to asymmetries in the polar vortex and, hence, maybe important in understanding sudden stratospheric warmings. The manuscript is worth to be published in Annales Geophysicae after some

minor changes/improvements.

Comments: The authors should include a table summarizing the different experiments. The reviewer found it a bit confusing that the Gaussian GW hotspot is shown in Figure 2 and the comparison between the 'used' simulations and the Gaussian is later presented in Figure 8. It is suggested to present this type of sensitivity in the experimental description section and later just refer to the 'main' model experiments.

Please clarify whether the altitudes shown in the Figures correspond to the pressure grid or whether the altitudes are computed from the geopotential and converted into a geometric altitude. This simplifies some comparisons to observations results.

Another comment of the reviewer concerns the implementation of the GW drag for the different latitudes. As the drag scales also with the atmospheric air mass that is affected by the drag, it might be mentioned that at higher latitudes essentially less drag is exerted to the atmosphere as the GW drag volume scales with latitude. Or with other words the atmospheric mass that is affected by the drag decreases with latitude. This might need some more discussion or should at least be mentioned in the interpretation of the results.

Some Figures (3,4,6 and 7) need an improvement of the quality.

Page 3: line 18: . . .vertical resolution 2842 km should be 2842 m

---

## Author Comment (AC1)

Response to the comments of reviewer #1,

we are thankful to the reviewer, whose comments helped us to improve the paper. We have revised the paper according to the remarks, and hope that we sufficiently responded to each concern.

**SC 1**: We already discussed most of the comments of reviewer #1 in detail and now shortly summarize most of the points.

In the following we are using the abbreviations **P** and **L** for page and line referring to the revised paper.

We attached the revised version of the paper. The red color refers to the comments of reviewer #1.

**In the paper the relevance of the work for the understanding of SSWs should be discussed (SSWs are not mentioned at all in the paper).**

We add a part about the preconditioning of the polar vortex in the revised version. Please also look at our response to comment no. 9.

**It should be qualitatively discussed how the model simulations agree with observations of gravity waves during SSWs.**

We observe a reduction of the of the GW flux, when the zonal mean zonal wind is decreasing at high latitudes like Ern et al. 2016 already reported. However, this effect is not very pronounced. We only observe a zonal mean zonal wind decrease of 10 ms$^{-1}$. There is no reversal of the zonal mean zonal wind conditions, which may lead to significant changes in the background strongly influencing the GW propagation (this is only the case during SSWs).
**P18L1-9**

**1. pg.2, line 2**
   **jet sources of gravity waves are not limited to frontal systems, and also the reference Plougonven et al. addresses jet sources of gravity waves more generally**

We replaced the expression 'frontal systems' by including 'jet sources'.
**P2L2**

**2. pg.2, line 5**
   **for the exponential increase with height constant background conditions have to be assumed and the gravity wave should propagate conservatively**

We included these conditions in the text.
**P2L5-6**

**3. pg.2, line 10**
   **The expression "filtered out" could be misleading in this context. The waves moving into the direction of the wind, but faster, are not necessarily completely removed from the spectrum. Their amplitude growth, however, is hampered by their slow intrinsic phase speed.**

We corrected it by adding that these GWs are partly filtered out when the phase speed of the GWs is equal to the background flow (mainly in the region of the mesospheric jet).
**P2L11**

**4. pg.2, line 10, suggestion:**
   **"For this reason the wind reverses in the mesosphere due to GW breaking, while in the opposite**

**direction to u travelling GWs deposit their momentum (Lindzen, 1981; Holton, 1982)."**
**"In mesosphere even GWs propagating in the opposite direction to u saturate and deposit their momentum. For this reason the wind reverses in the upper mesosphere/lower thermosphere (Lindzen, 1981; Holton, 1982)."**

Indeed our phrasing was a bit misleading. We included your new version into the revised paper.
**P2L11-13**

**5. pg.2, line 20/21**
**Compared to the enumeration of mountain wave sources, the enumeration of convective regions generating gravity waves is quite short. Suggestion for keeping the balance:**
**"Typically, in the stratosphere satellite observations show a characteristic structure of enhanced GW activity in the subtropics that is caused by deep convection over Southeast Asia, America, Africa, or the Maritime Continent in the respective summer season (Jiang et al., 2004; Ern and Preusse, 2012; Wright and Gille, 2011)."**

Because our observed Asian GW hotspot is partly convectively and orographically generated we should represent both trigger mechanisms (convection and orography) equally. We included your new version into the revised paper.
**P2L22-24**

**6. pg.2, line 22**
**Here you write "...objective determination of the GW drag from satellite measurements alone is not possible."**
**This is a too strong statement that also does not hold. Estimates based on satellite data exist, however contain large errors.**
**In the text following line 22 you give two references and state that GW drag would be derived from superpressure balloons and lidar data. However, these two references are used out-of-context!**
**The GW drag in these papers is based on model simulations alone. So far GW drag has not been derived from superpressure balloons because these balloons float on a fixed altitude, and vertical gradients cannot be inferred. To my knowledge, GW drag has not been tried from lidar data (please correct me if I am wrong), but I know that GW drag has been derived from radar data. Similar as GW drag derived from satellite, these estimates generally have large errors. Therefore I would suggest the following rewording:**
**"Reliable estimates of GW drag from observations are generally difficult. There are attempts from satellite (for example, Ern et al., 2014; Ern et al., 2016), or from radar (for example, Reid and Vincent, 1987). Usually, however, uncertainties of these estimates are quite large. From model studies there are indications that GWs can break already in the lower stratosphere (LS), for example Plougonven et al. (2008), or Constantino et al. (2015)."**

As you said, it is possible to derive the drag by making different assumptions, which create large errors. We changed this part partly based on your suggested text.
**P2L24-27**

**7. pg.3, line 29**
**This comment is not relevant for the current paper, but may become relevant for future work. On pg.3 line 29 you write that GW potential energy of the MUAM model would be tuned "... based on potential energy data obtained from GPS radio occultation measurements" Tuning the model towards GW potential energies observed from GPS may introduce large biases. It has been shown by Rapp et al. (2018) that vertical filtering of GPS soundings for obtaining the gravity wave signal does not remove larger scale structures having short vertical wavelengths, such as inertial instabilities at low and mid latitudes, or Kelvin waves in the tropics (Ern et al., 2008).**

We shortly discussed the advantages and disadvantages by using GPS RO data for the GW initialization within the model.
**P4L5-10**

**8. pg.15, line 30**
   **Gravity wave hotspots do not only occur over mountain ridges. Local hotspots can also be caused by the GW jet source mechanism. For example, Ern et al., 2016 find enhancements of GW activity that are linked to jet exit regions. Possibly these hotspots are missed by AIRS because of its limitation to gravity waves of very long vertical wavelengths. The proposed follow-up study is therefore a promising way to proceed, but should not be limited to the positions of orographic GW hotspots. Also longitudinally varying positions of jet-related GW sources could be important.**

We included this type of GW as possible contributor of the artificial GW forcing in our conclusion.
**P18L16-17**

**9. General comment for the discussion throughout the paper:**
   **For midlatitude forcing some of the findings are very similar to the situation during SSWs.**
   **- vortex slowdown and shift (Fig. 3)**
   **- stratospheric warming (Fig. 4)**
   **- changes in the activity of the SPW1**
   **These similarities should be discussed in more detail and put into the context of SSWs. For example, it has been suggested recently by Albers and Birner (2014) that gravity wave forcing at midlatitudes could be important for the onset of SSWs. Also satellite observations of gravity waves show stronger GW drag at mid or even low latitudes prior to major SSW central dates (Ern et al., 2016). In the same paper strong gravity wave activity and GW drag was seen in the polar vortex during 2011. In this year, however, the vortex was very stable and confined to high latitudes which also confined gravity wave activity to high latitudes, apparently without affecting the stability of the vortex. This supports one of the main findings of your study and should therefore be mentioned.**

Based on our results showing a slight warming of the polar stratosphere as well as a decreasing zonal mean zonal wind at high latitudes (slight displacement of the polar vortex) we included this topic (preconditioning of the polar vortex) in our introduction and discussion.
**P2L28-32, P13L16-20, P17L25-34**

**Other Comments**
**1. pg.2, lline 28/29: Please rewrite the following sentence for better legibility!**
   **"This study is focusing on different GW breaking areas in the lower stratosphere along and the effects on the middle atmosphere highly depending on the position."**

We rewrote this part.
**P2L33-34**

**2. pg.2, line 29: who were concentrating ! who were focusing**

Done.

**3. pg.3, line 2: shifting meridionally the EA/NP hotspot along its fixed longitude range! shifting meridionally the EA/NP hotspot keeping its longitude range fixed**

Done.

**4. pg.3, line 23: interval, in which ! interval in which**

Done.

**5. pg.4, line 3: refer to the mid ! referring to the mid**

Done.

**6. pg.4, line 17: reproduce ! reproduces**

Done.

**7. pg.5, line 2: observations but the ! observations, but the**

Done.

**8. pg.5, line 3: jet filtering some of ! jet filtering of some of**

Done.

**9. pg.7, line 4: is shifted toward ! that is shifted toward**

Done.

**10. pg.7, line 14: kmat ! km at**

Done.

**11. pg.8, line 14: west wind ! westerly wind**

Done.

**12. pg.11, line 10: more SPW 1 are excited, ! SPW 1 excitation is strengthened,**

Done.

**13. pg.12, line 2: fluxes and its ! fluxes and their**

Done.

**14. pg.13, line 3: atmopshere. ! atmosphere.**

Done.

[revised manuscript text omitted]

10 usually observed in the middle atmosphere. Also GWs being faster than the background wind are able to propagate but they are mostly filtered out by the strong polar-night jet at the latest, when $c$ becomes equal to $u$. In the mesosphere GWs, which are propagating in the opposite direction to $u$ saturate and deposit their momentum. For this reason the wind reverses in the upper mesosphere/lower thermosphere (MLT) (Lindzen, 1981; Holton, 1982). The transfer of energy and momentum by breaking GWs is also called GW drag.

15 Owing to the variety of sources GWs have a large spatial and temporal variability. To capture the global distribution of GWs, the potential energy ($E_{pot}$), momentum flux (MF), or stability indicators (Pišoft et al., 2018) can be estimated by using satellite data (Ern et al., 2004; Fröhlich et al., 2007; Hoffmann et al., 2013; Schmidt et al., 2016). These numerous observational studies highlight a number of different local GW hotspots, which are mainly generated by orography and convection. The most common GW hotspots are the orographically induced GW hotspots near the Alps (Hierro et al., 2018), the Andes (Llamedo et al.,

20 2009; Alexander et al., 2010; Lilienthal et al., 2017), the Antarctic Peninsula (Moffat-Griffin et al., 2010), the Himalayas (Kumar et al., 2012), the Mongolian Plateau (White et al., 2018), the Rocky Mountains (Lilly et al., 1982) and in the Scandinavian region (Kirkwood et al., 2010). Typically, satellite observations show a characteristic structure of enhanced GW activity in the subtropical stratosphere that is caused by deep convection over Southeast Asia, America, Africa, or the Maritime Continent in the respective summer season (Jiang et al., 2004; Wright and Gille, 2011; Ern and Preusse, 2012). However, reliable estimates

25 of GW drag from observations are generally difficult. Several methods have been established to derive the GW drag from satellite (e.g. Ern et al., 2014, 2016), or from radar measurements (e.g. Reid and Vincent, 1987). However, uncertainties of these estimates are quite large.

From model studies there are indications that GWs can already break in the lower stratosphere (LS) (e.g. Plougonven et al., 2008; Constantino et al., 2015), which leads to an additional transfer of momentum and energy in this region. In connection

30 with high PW activity this is highly affecting the stability of the polar vortex and can cause a sudden stratospheric warming (SSW) (Albers and Birner, 2014). This effect has been also observed in satellite measurements showing enhanced GW drag before SSWs (Ern et al., 2016). Thus, an additional GW forcing may lead to a preconditioning of the polar vortex.

This study is focusing on the role of the zonal position of localized GW breaking areas and their effects on the middle atmosphere dynamics. It is motivated by the findings of Šácha et al. (2015) who were focusing on the East Asian/North Pacific

[revised manuscript text omitted]

Their analysis time period was much shorter and the declination of the sun was different. They also nudged the model zonal mean temperature up to 30 km. In this regard, our experimental set-up might be considered superior to their simulations, especially, because the nudging does not interfere with the implemented GW forcing in this new configuration. We refer to this reference simulation as the 'Ref' simulation.

[revised manuscript text omitted]

The H3 mean value (mean $GWD_u$: -10 ms$^{-1}$day$^{-1}$) is roughly 3300 larger than the one of the Ref simulation (mean $GWD_u$: 0.003 ms$^{-1}$day$^{-1}$) within the region of the EA/NP hotspot. These maximum values of the $GWD_u$ as well as those of the $GWD_v$ and the $GWD_T$ are summarized in Tab. 1 for the Ref and the GW hotspot simulations. In spite of the huge difference compared to the Ref simulation, the zonal GW forcing is moderate in terms of what is estimated from observations (40 ms$^{-1}$day$^{-1}$ and more) and from GW parameterizations in this region (Šácha et al., 2018). Concerning the meridional GW drag and the heating due to breaking GWs the maximum (mean) value of the H3 simulation is only 5 (100) times larger than the one of the Ref simulation (not shown here). To investigate possible effects with regard to the position of the GW hotspot we performed a sensitivity study. For this, we kept the longitude (118.1°E-174.3°E) and altitude (18-30 km) range as well as the zonal extent of 25° fixed, but varied the observed GW hotspot in 5° steps from 27.5°N-52.5°N (simulation H1) to 62.5°N-87.5°N (simulation H8), while labeling the experiments inbetween by H2 through H7 (see Tab. 1).

To analyze possible effects of the sharp transition zone between the unchanged and enhanced GW drag, additional simulations with a smoothed GW forcing were performed, by using a 3D Gaussian function, with a standard deviation of 10°, 22.5°, and 5.684 km in the zonal, meridional, and vertical direction, respectively. To get the same integral forcing as in the H1-H8 simulations, the size or the intensity of the local GW forcing as Gaussian distribution needs to be adjusted. For our experiments, we mainly increased the strength of the local GW forcing and only slightly increased the size. The maximum values for the $GWD_u$, $GWD_v$, and $GWD_T$ forcing as 3D Gaussian distribution have been chosen as -13 ms$^{-1}$day$^{-1}$, -0.13 ms$^{-1}$day$^{-1}$, and 0.065 Kday$^{-1}$ (see Tab. 1), respectively. The 3D Gaussian distribution for the H3 GW hotspot can be seen in the right panel of Fig. 2. In this paper we mainly concentrate on the 3D GW hotspots shaped like a box, when we analyse the effects on the middle atmosphere dynamics. For comparison regarding the shape of the artificial GW forcing, we just focus on the H3 GW hotspot with Gaussian smoothed boundaries, when we discuss the SPW modulation in section 3.2, which may be affected by the GW hotspots with sharp boundaries.

By comparing the size of the GW hotspots it is obvious (can be seen in Fig. 5) that the area of the enhanced GW drag, which scales with the cosine of the latitude, is decreasing for increasing latitude. But scaling the GW drag with latitude would lead to a much larger zonal mean GW drag at high latitudes and resulting changes in the circulation. Also the horizontal winds, which are affected by resulting nonlinear interactions are scaled in the model equations. In the current approach we are conserving the ratio of enhanced and unchanged GW drag values within the respective latitudinal belt, which is more meaningful. Also, the horizontal wavelength of PWs is smaller the closer to the pole they are, so that the ratio of the width of the GW forcing and the horizontal wavelength of the PWs remains the same for the respective latitudinal belt. In the following we will show that the spatial shape as well as the spatial size of the local GW forcing is not the most decisive factor, when we are comparing the 3D Gaussian distribution with the 3D GW forcing shaped as a box. Thus, GW hotspots having all the same size may lead to comparable results.

[revised manuscript text omitted]

To sum up the influence on the polar vortex, in section 3.1 we have observed a slight warming of the lower stratosphere and the decreasing west wind at middle to high latitudes, which indicates a weakening of the polar vortex as a consequence of the GW drag enhancement. Thus, the stability of the polar vortex is not only depending on the PW activity but rather on the interplay or nonlinear interaction of GW and PW forcings. The anomalous SPW forcing and suppression of SPW propagation show that

20 the GW drag can play an important role for the preconditioning of the polar vortex (see next section).

[revised manuscript text omitted]

Based on these results of the sensitivity study we have seen that a local GW forcing can lead to a weakening (warming of the lower stratosphere and slight displacement) of the polar vortex at high latitudes, which is highly depending on the strength (Šácha et al., 2016) as well as on the zonal distribution of the forcing (this study). Usually, it is assumed that the preconditioning of the polar vortex is mainly driven by enhanced PW activity (Labitzke, 1981). But there are also several indications based on satellite observations (Ern et al., 2016) and reanalysis data (Albers and Birner, 2014) showing that the GW drag and the absolute GW momentum flux is enhanced (reduced) in the stratosphere right before (after) SSWs. Albers and Birner (2014) have analyzed the total wave forcing from Japanese Meteorological Agency and Central Research Institute of Electrical Power Industry 25-year Reanalysis (JRA-25) data before SSW events and found that up to 70% of the total drag is induced by orographic GWs. Ern et al. (2016) directly derived the GW drag and absolute momentum fluxes from HIRDLS and SABER temperatures and found out that both parameters are enhanced before and around the central day of a SSW (strong polar jet)

and reduced, when the zonal wind is weak (after SSW). Because we kept the GW drag forcing constant during the whole experiments we cannot evaluate nonlinear effects, which would possibly reduce the GW drag connected with the displacement of the polar vortex. Furthermore, we have a fixed GW source distribution so that no additional GWs are generated owing to changes in the tropospheric circulation. But on the basis of the zonal and meridional GW flux, which is changing according to the propagation condtions, we have analyzed the absolute horizontal GW momentum flux (not shown here). In this case, we observe as well a reduction of the GW flux, when the zonal mean zonal wind is decreasing at high latitudes. However, this effect is not very pronounced in our experiments because the zonal mean zonal wind differences are much smaller than during a real SSW event. We only observe zonal mean zonal wind differences of about -10 ms$^{-1}$, which do not lead to a reversal of the zonal mean zonal wind, and thus, to significant background changes, which may strongly influence the GW propagation.

[revised manuscript text omitted]

---

## Author Comment (AC2)

Response to the comments of reviewer #2, we are thankful to the reviewer, whose comments helped us to improve the paper. We have revised the paper according to the remarks, and hope that we sufficiently responded to each concern.

**SC 2**: We already discussed most of the comments of reviewer #2 in detail and now shortly summarize most of the points.

In the following we are using the abbreviations **P** and **L** for page and line referring to the revised paper.

We attached the revised version of the paper. The blue color refers to the comments of reviewer #2.

**The authors should include a table summarizing the different experiments.**

We insert a table including the position as well as the positive and negative maxima of the zonal and meridional GW drag and heating by GWs.
**P6**

**The reviewer found it a bit confusing that the Gaussian GW hotspot is shown in Figure 2 and the comparison between the 'used' simulations and the Gaussian is later presented in Figure 8. It is suggested to present this type of sensitivity in the experimental description section and later just refer to the 'main' model experiments.**

We added more explanations to the description of Figure 2 and the new Table 1, and clearly stated that we will come back to the Gaussian-shaped hotspot later.
**P7L2-3, P7L17-21**

**Please clarify whether the altitudes shown in the Figures correspond to the pressure grid or whether the altitudes are computed from the geopotential and converted into a geometric altitude. This simplifies some comparisons to observations results.**

We added an explanation of the logarithmic pressure height in the revised paper.
**P3L18-22**

**Another comment of the reviewer concerns the implementation of the GW drag for the different latitudes. As the drag scales also with the atmospheric air mass that is affected by the drag, it might be mentioned that at higher latitudes essentially less drag is exerted to the atmosphere as the GW drag volume scales with latitude. Or with other words the atmospheric mass that is affected by the drag decreases with latitude. This might need some more discussion or should at least be mentioned in the interpretation of the results.**

We added a note on this in section 2 and listed some arguments explaining why we did not weight the artificial GW forcing with the latitude.
**P7L22-31**

**Some Figures (3,4,6 and 7) need an improvement of the quality.**

We improved the resolution of these figures.
**P8, P9, P11, P12**

**Page 3: line 18: . . .vertical resolution 2842 km should be 2842 m**

We are using the '.' for the separation of decimal numbers.

[revised manuscript text omitted]

2009; Alexander et al., 2010; Lilienthal et al., 2017), the Antarctic Peninsula (Moffat-Griffin et al., 2010), the Himalayas (Kumar et al., 2012), the Mongolian Plateau (White et al., 2018), the Rocky Mountains (Lilly et al., 1982) and in the Scandinavian region (Kirkwood et al., 2010). Typically, satellite observations show a characteristic structure of enhanced GW activity in the subtropical stratosphere that is caused by deep convection over Southeast Asia, America, Africa, or the Maritime Continent in the respective summer season (Jiang et al., 2004; Wright and Gille, 2011; Ern and Preusse, 2012). However, reliable estimates of GW drag from observations are generally difficult. Several methods have been established to derive the GW drag from satellite (e.g. Ern et al., 2014, 2016), or from radar measurements (e.g. Reid and Vincent, 1987). However, uncertainties of these estimates are quite large.

From model studies there are indications that GWs can already break in the lower stratosphere (LS) (e.g. Plougonven et al., 2008; Constantino et al., 2015), which leads to an additional transfer of momentum and energy in this region. In connection with high PW activity this is highly affecting the stability of the polar vortex and can cause a sudden stratospheric warming (SSW) (Albers and Birner, 2014). This effect has been also observed in satellite measurements showing enhanced GW drag before SSWs (Ern et al., 2016). Thus, an additional GW forcing may lead to a preconditioning of the polar vortex.

[revised manuscript text omitted]

Shved (1985).

GWs are parameterized after an updated linear scheme (Lindzen, 1981; Jakobs et al., 1986) with multiple breaking levels (Fröhlich et al., 2003b; Jacobi et al., 2006). GW amplitudes are included at an altitude of 10 km as zonal mean with a global average of $1 \, \mathrm{cms}^{-1}$ for the vertical velocity perturbation. This value is weighted by a prescribed zonal mean GW amplitude distribution based on $E_{pot}$ data obtained from GPS radio occultation measurements (Šácha et al., 2015; Lilienthal et al., 2017). Although the $E_{pot}$ data still contain Kelvin waves and other possible wave structures with short vertical wavelengths, which may introduce biases, the GW amplitude distribution is more realistic compared to the hyperbolic tangent function of the latitude, which was used in earlier experiments (Jacobi et al., 2006) and leads to an improvement of the zonal mean GW climatology. It shows maximum GW amplitudes (not shown here) at the equator (convectively generated GWs) and at midlatitudes (orographically induced GWs). At each grid point 48 waves are induced propagating in eight different directions with six different phase speeds ranging from 5 to $30 \, \mathrm{ms}^{-1}$.

In this configuration based on January decadal mean (2000-2010) ERA Interim reanalysis data we create a reference simulation with a spin-up period of 270 days, in which the mean circulation is built up and different waves like planetary waves (PWs) and tides are generated. The declination and the ozone and carbon dioxide concentration are fixed to avoid further non-zonal structures being induced besides to the enhanced GW forcing. The declination corresponds to January 15 (referring to the mid of the month) and the ozone and carbon dioxide data are taken from the year 2005 (refer to the mid of the decade). For the analysis, a time interval of 120 days with a temporal resolution of 2 hours after the spin-up period was modeled. Šácha et al. (2016) have already analysed the effect of the Asian hotspot with MUAM by performing a sensitivity study with regard to the strength of the GW forcing in the stratosphere.

[Figure]

**Figure 1.** January zonal and monthly mean of the (a) zonal wind ($\mathrm{ms}^{-1}$), (b) meridional wind ($\mathrm{ms}^{-1}$), (c) temperature (K), (d) zonal GW fluxes ($\mathrm{m}^2\mathrm{s}^{-2}$), (e) zonal wind acceleration through breaking GWs ($\mathrm{ms}^{-1}\mathrm{day}^{-1}$) and (f) SPW 1 amplitude ($\mathrm{ms}^{-1}$) extracted from the zonal wind of the reference simulation.

Their analysis time period was much shorter and the declination of the sun was different. They also nudged the model zonal mean temperature up to 30 km. In this regard, our experimental set-up might be considered superior to their simulations, especially, because the nudging does not interfere with the implemented GW forcing in this new configuration. We refer to this reference simulation as the 'Ref' simulation.

[revised manuscript text omitted]

The H3 mean value (mean GWD$_u$: -10 ms$^{-1}$day$^{-1}$) is roughly 3300 larger than the one of the Ref simulation (mean GWD$_u$: 0.003 ms$^{-1}$day$^{-1}$) within the region of the EA/NP hotspot. These maximum values of the GWD$_u$ as well as those of the GWD$_v$ and the GWD$_T$ are summarized in Tab. 1 for the Ref and the GW hotspot simulations. In spite of the huge difference compared to the Ref simulation, the zonal GW forcing is moderate in terms of what is estimated from observations (40 ms$^{-1}$day$^{-1}$ and more) and from GW parameterizations in this region (Šácha et al., 2018). Concerning the meridional GW drag and the heating due to breaking GWs the maximum (mean) value of the H3 simulation is only 5 (100) times larger than the one of the Ref simulation (not shown here). To investigate possible effects with regard to the position of the GW hotspot we performed a sensitivity study. For this, we kept the longitude (118.1°E-174.3°E) and altitude (18-30 km) range as well as the zonal extent of 25° fixed, but varied the observed GW hotspot in 5° steps from 27.5°N-52.5°N (simulation H1) to 62.5°N-87.5°N (simulation H8), while labeling the experiments inbetween by H2 through H7 (see Tab. 1).

To analyze possible effects of the sharp transition zone between the unchanged and enhanced GW drag, additional simulations with a smoothed GW forcing were performed, by using a 3D Gaussian function, with a standard deviation of 10°, 22.5°, and 5.684 km in the zonal, meridional, and vertical direction, respectively. To get the same integral forcing as in the H1-H8 simulations, the size or the intensity of the local GW forcing as Gaussian distribution needs to be adjusted. For our experiments, we mainly increased the strength of the local GW forcing and only slightly increased the size. The maximum values for the GWD$_u$, GWD$_v$, and GWD$_T$ forcing as 3D Gaussian distribution have been chosen as -13 ms$^{-1}$day$^{-1}$, -0.13 ms$^{-1}$day$^{-1}$, and 0.065 Kday$^{-1}$ (see Tab. 1), respectively. The 3D Gaussian distribution for the H3 GW hotspot can be seen in the right panel of Fig. 2. In this paper we mainly concentrate on the 3D GW hotspots shaped like a box, when we analyse the effects on the middle atmosphere dynamics. For comparison regarding the shape of the artificial GW forcing, we just focus on the H3 GW hotspot with Gaussian smoothed boundaries, when we discuss the SPW modulation in section 3.2, which may be affected by the GW hotspots with sharp boundaries.

By comparing the size of the GW hotspots it is obvious (can be seen in Fig. 5) that the area of the enhanced GW drag, which scales with the cosine of the latitude, is decreasing for increasing latitude. But scaling the GW drag with latitude would lead to a much larger zonal mean GW drag at high latitudes and resulting changes in the circulation. Also the horizontal winds, which are affected by resulting nonlinear interactions are scaled in the model equations. In the current approach we are conserving the ratio of enhanced and unchanged GW drag values within the respective latitudinal belt, which is more meaningful. Also, the horizontal wavelength of PWs is smaller the closer to the pole they are, so that the ratio of the width of the GW forcing and the horizontal wavelength of the PWs remains the same for the respective latitudinal belt. In the following we will show that the spatial shape as well as the spatial size of the local GW forcing is not the most decisive factor, when we are comparing the 3D Gaussian distribution with the 3D GW forcing shaped as a box. Thus, GW hotspots having all the same size may lead to comparable results.

[revised manuscript text omitted]

To sum up the influence on the polar vortex, in section 3.1 we have observed a slight warming of the lower stratosphere and the decreasing west wind at middle to high latitudes, which indicates a weakening of the polar vortex as a consequence of the GW drag enhancement. Thus, the stability of the polar vortex is not only depending on the PW activity but rather on the interplay or nonlinear interaction of GW and PW forcings. The anomalous SPW forcing and suppression of SPW propagation show that the GW drag can play an important role for the preconditioning of the polar vortex (see next section).

[revised manuscript text omitted]

Based on these results of the sensitivity study we have seen that a local GW forcing can lead to a weakening (warming of the lower stratosphere and slight displacement) of the polar vortex at high latitudes, which is highly depending on the strength (Šácha et al., 2016) as well as on the zonal distribution of the forcing (this study). Usually, it is assumed that the preconditioning of the polar vortex is mainly driven by enhanced PW activity (Labitzke, 1981). But there are also several indications based on satellite observations (Ern et al., 2016) and reanalysis data (Albers and Birner, 2014) showing that the GW drag and the absolute GW momentum flux is enhanced (reduced) in the stratosphere right before (after) SSWs. Albers and Birner (2014) have analyzed the total wave forcing from Japanese Meteorological Agency and Central Research Institute of Electrical Power Industry 25-year Reanalysis (JRA-25) data before SSW events and found that up to 70% of the total drag is induced by orographic GWs. Ern et al. (2016) directly derived the GW drag and absolute momentum fluxes from HIRDLS and SABER temperatures and found out that both parameters are enhanced before and around the central day of a SSW (strong polar jet)

and reduced, when the zonal wind is weak (after SSW). Because we kept the GW drag forcing constant during the whole experiments we cannot evaluate nonlinear effects, which would possibly reduce the GW drag connected with the displacement of the polar vortex. Furthermore, we have a fixed GW source distribution so that no additional GWs are generated owing to changes in the tropospheric circulation. But on the basis of the zonal and meridional GW flux, which is changing according to the propagation condtions, we have analyzed the absolute horizontal GW momentum flux (not shown here). In this case, we observe as well a reduction of the GW flux, when the zonal mean zonal wind is decreasing at high latitudes. However, this effect is not very pronounced in our experiments because the zonal mean zonal wind differences are much smaller than during a real SSW event. We only observe zonal mean zonal wind differences of about -10 ms$^{-1}$, which do not lead to a reversal of the zonal mean zonal wind, and thus, to significant background changes, which may strongly influence the GW propagation.

Another interesting aspect is that different shapes of the local GW forcing will not have strong effects on the circulation. Despite of Gaussian-smoothed boundaries just negligible changes can be observed in the dynamics and SPW development, which are mainly due to the varying GW drag in the three dimensional Gaussian distribution leading to larger (smaller) effects when the Gaussian distribution is maximizing (minimizing).

Comparing the positions of these simulated GW hotspots with measurements (e.g. Hoffmann et al., 2013) it is clear that at least some of the latitudinally shifted GW hotspots are not very realistic so that our experiments should only be considered as a qualitative sensitivity study. Regarding orography there are no obvious sources, when we displace the GW hotspot latitudinally. But indeed, some of these GW hotspots can be purely hypothetical connected to jet exit regions. To make it more realistic, the next step will be to analyse the effect of a longitudinally shifted hotspot (fixed latitude range between 30°N and 60°N), because observations and GCM experiments have shown their existence (listed in the introduction). In this latitude range GW

hotpots like the Himalayan region, the Alps or the Rocky mountains are part of the experiments. Also the interaction of two or more GW hotspots is part of our interests and will provide more insights into the effect of a localized GW forcing, which may be also important for the development of new GW parametrizations.

*Code availability.* MUAM model code is available from the corresponding author upon request.

*Author contributions.* N. Samtleben performed the MUAM model simulations and drafted the first version of the manuscript. P. Šácha provided GW potential energy distributions. P. Šácha, P. Pišoft, A. Kuchař and C. Jacobi actively contributed to the discussions and the writing of the paper.

*Competing interests.* C. Jacobi is one of the Editors-in-Chief of Annales Geophysicae. The authors declare that there are no conflicts of interest.

*Acknowledgements.* This study has been supported by Deutsche Forschungsgemeinschaft (DFG) under grant JA836/32-1, and by GA CR under grant 16-01562J. ERA Interim reanalysis data have been provided by ECMWF through apps.ecmwf.int/datasets/data/. Petr Pišoft and Aleš Kuchař were supported by GA CR under grant nos. 16-01562J and 18-01625S. Petr Šácha was supported by GA CR under grant nos. 16-01562J and 18-01625S and by Government of Spain under grant no. CGL2015-71575-P and in later stages of the manuscript preparation through a postdoctoral grant of the Xunta de Galicia ED481B 2018/103. The EPhysLab is supported through the European Regional Development Fund (ERDF).

[revised manuscript text omitted]

---

## Author Response (AR1)

We are thankful for your comments, which helped us to improve the paper. We have revised the paper according to the remarks, and hope that we sufficiently responded to each concern.

In the following we are using the abbreviations **P** and **L** for page and line.

We attached the revised version of the paper.

**Comments of Reviewer 1:**

**In the paper the relevance of the work for the understanding of SSWs should be discussed (SSWs are not mentioned at all in the paper).**

Yes, we did this in the revised version. Please also look at our response to comment no. 9.

**It should be qualitatively discussed how the model simulations agree with observations of gravity waves during SSWs.**

This is a good point, but it is hard to realize because we kept nearly all the conditions constant during our experiments. We save the GW drag data from the reference simulation, and then increased the GW drag in the specific region, and finally put it back into the model. We chose this cumbersome way, because otherwise we would get feedback mechanism which would in turn change the GW drag distribution. It was our intention to directly see the impact of this hotspot being not influenced by nonlinear effects. So, on the basis of the GW drag we do not see any changes. What is still changing is the horizontal GW momentum flux, which can be partly analyzed. It is just changing because of changes in the propagation conditions (background changes). The GW sources are fixed in MUAM thus, no additional GWs are generated. Ern et al. [2016] presented satellite measurements showing the absolute GW momentum flux during several years, also including SSWs. In general, they found out that the absolute GW momentum flux is increased (i) when the polar jet is strong and (ii) before and around the central day of a SSW and (iii) it is reduced when the zonal wind is weak. We tried to compare this to our GW output data containing the zonal and meridional GW flux, which we used to calculate the zonal mean absolute horizontal GW flux averaged between 60-80°N. It is scaled by density and presented in Fig. 1 in a time-height plot to show the temporal development. In contour lines the zonal mean zonal wind also averaged between 60 and 80°N is shown. The results in Fig. 1 are based on the H3 simulation (observed Asian GW hotspot). During the first 10 to 20 days the zonal mean zonal wind is decelerating, and the absolute horizontal GW flux decreases up to 70 km, which would correspond to the results from Ern et al. [2016]. However, in our simulation this effect is not very pronounced. We discussed this in the revised version.

[Figure]

Figure 1: Zonal mean absolute horizontal GW flux scaled by density (color coding) and the zonal mean zonal wind (contour lines) in a time-height plot during the analysed 120 model days. The data is based on the H3 simulation (Asian GW hotspot).

**P18L16-24**

**1. pg.2, line 2**
   jet sources of gravity waves are not limited to frontal systems, and also the reference
   Plougonven et al. addresses jet sources of gravity waves more generally

We agree that 'frontal systems' is too specific in this case. It is one source generating GWs owing to adjustment processes. Thus, we included jet sources.
**P2L2**

**2. pg.2, line 5**
   for the exponential increase with height constant background conditions have to be assumed
   and the gravity wave should propagate conservatively

We included these conditions in the text.
**P2L5-6**

**3. pg.2, line 10**
   The expression "filtered out" could be misleading in this context. The waves moving into the
   direction of the wind, but faster, are not necessarily completely removed from the spectrum.
   Their amplitude growth, however, is hampered by their slow intrinsic phase speed.

This is an infelicity of the expression at this point. When they are faster than the background then the GWs are of course not filtered out but if they reach the critical line (background flow equal to the phase speed – for most of the GWs at the latest in the altitude of the mesospheric jet) the GWs are absorbed by the background flow or reflected back to the troposphere. We corrected it by adding that these GWs are partly filtered out when the phase speed of the GWs is equal to the background flow (mainly in the region of the mesospheric jet).
**P2L11-12**

**4. pg.2, line 10, suggestion:**
   "For this reason the wind reverses in the mesosphere due to GW breaking, while in the opposite
   direction to u travelling GWs deposit their momentum (Lindzen, 1981; Holton, 1982)."
   "In mesosphere even GWs propagating in the opposite direction to u saturate and deposit their
   momentum. For this reason the wind reverses in the upper mesosphere/lower thermosphere
   (Lindzen, 1981; Holton, 1982)."

Thank you for this suggestion. Our sentence contained a huge amount of information and was quite confusing wherefore your version is much better and we included this one in our paper.
**P2L12-14**

**5. pg.2, line 20/21**
   Compared to the enumeration of mountain wave sources, the enumeration of convective
   regions generating gravity waves is quite short. Suggestion for keeping the balance:
   "Typically, in the stratosphere satellite observations show a characteristic structure of enhanced
   GW activity in the subtropics that is caused by deep convection over Southeast Asia, America,
   Africa, or the Maritime Continent in the respective summer season (Jiang et al., 2004; Ern and
   Preusse, 2012; Wright and Gille, 2011)."

Thank you for this suggestions. This is a good idea to keep them in balance. Also because our observed Asian GW hotspot is partly convectively and orographically generated so that we should represent both trigger mechanisms (convection and orography) equally.
**P2L25-27**

**6. pg.2, line 22**
Here you write "...objective determination of the GW drag from satellite measurements alone is not possible."
This is a too strong statement that also does not hold. Estimates based on satellite data exist, however contain large errors.
In the text following line 22 you give two references and state that GW drag would be derived from superpressure balloons and lidar data. However, these two references are used out-of-context!
The GW drag in these papers is based on model simulations alone. So far GW drag has not been derived from superpressure balloons because these balloons float on a fixed altitude, and vertical gradients cannot be inferred. To my knowledge, GW drag has not been tried from lidar data (please correct me if I am wrong), but I know that GW drag has been derived from radar data. Similar as GW drag derived from satellite, these estimates generally have large errors. Therefore I would suggest the following rewording:
"Reliable estimates of GW drag from observations are generally difficult. There are attempts from satellite (for example, Ern et al., 2014; Ern et al., 2016), or from radar (for example, Reid and Vincent, 1987). Usually, however, uncertainties of these estimates are quite large. From model studies there are indications that GWs can break already in the lower stratosphere (LS), for example Plougonven et al. (2008), or Constantino et al. (2015)."

We agree with your point of view. Maybe our statement was a bit misleading because we meant the measurement of the GW drag itself. As you said, it is possible to derive the drag by making different assumptions which create large errors. We changed this part based on your suggested text.
**P2L28-35**

**7. pg.3, line 29**
This comment is not relevant for the current paper, but may become relevant for future work.
On pg.3 line 29 you write that GW potential energy of the MUAM model would be tuned "... based on potential energy data obtained from GPS radio occultation measurements" Tuning the model towards GW potential energies observed from GPS may introduce large biases. It has been shown by Rapp et al. (2018) that vertical filtering of GPS soundings for obtaining the gravity wave signal does not remove larger scale structures having short vertical wavelengths, such as inertial instabilities at low and mid latitudes, or Kelvin waves in the tropics (Ern et al., 2008).

Yes, that's true. This is something we kept in mind when we were analyzing the GPS RO data which as well do not consider Kelvin waves, for example. Before we included the globally weighted potential energy data as zonal mean into our model, we had an artificial zonal mean GW amplitude distribution based on a hyperbolic tangent function depending on the latitude which was less realistic then the one derived from the GPS RO data. We thought it is a good idea to replace it but of course we should not forget that these data does not include all small-scale disturbances. We discussed this further in the revised version.
**P4L15-19**

**8. pg.15, line 30**
Gravity wave hotspots do not only occur over mountain ridges. Local hotspots can also be caused by the GW jet source mechanism. For example, Ern et al., 2016 find enhancements of GW activity that are linked to jet exit regions. Possibly these hotspots are missed by AIRS because of its limitation to gravity waves of very long vertical wavelengths. The proposed follow-up study is therefore a promising way to proceed, but should not be limited to the positions of orographic GW hotspots. Also longitudinally varying positions of jet-related GW sources could be important.

Thank you for this suggestion. In this paper the position of the analyzed GW hotspots does not explicitly refer to a specific source. Regarding orography there are no obvious sources, when we are displacing

the GW hotspot latitudinally. But of course, some of these GW hotspots can be purely hypothetical connected to jet exit regions.

For this purpose, we are planning to write another paper about the longitudinal displacement of the GW hotspot. When we are displacing the GW hotspot longitudinally then we do not only capture/represent orographically generated GWs. Because this displacement is partly along the polar front jet, which may generate GWs we indirectly consider this kind of GWs. Thus, we also included this type of GW as possible contributor of the artificial GW forcing.
**P18L31-32**

**9. General comment for the discussion throughout the paper:**
   **For midlatitude forcing some of the findings are very similar to the situation during SSWs.**
   **- vortex slowdown and shift (Fig. 3)**
   **- stratospheric warming (Fig. 4)**
   **- changes in the activity of the SPW1**
   **These similarities should be discussed in more detail and put into the context of SSWs. For example, it has been suggested recently by Albers and Birner (2014) that gravity wave forcing at midlatitudes could be important for the onset of SSWs. Also satellite observations of gravity waves show stronger GW drag at mid or even low latitudes prior to major SSW central dates (Ern et al., 2016). In the same paper strong gravity wave activity and GW drag was seen in the polar vortex during 2011. In this year, however, the vortex was very stable and confined to high latitudes which also confined gravity wave activity to high latitudes, apparently without affecting the stability of the vortex. This supports one of the main findings of your study and should therefore be mentioned.**

Thank you for this comment! Indeed, the results show specific characteristics, which are comparable to SSWs. In this case, we did not create a SSW but caused a preconditioning of the polar vortex owing to the aspects (SPW activity, displacement and slight warming) you already mentioned. But this also strongly depends on the strength of the forcing (stronger forcing can also lead to a total breakdown of the polar vortex). Based on this fact we included this topic (preconditioning of the polar vortex) in our discussion and referred to this topic in the introduction.
**P3L2-5, P13L6-10, P18L6-16**

**Other Comments**
**1. pg.2, lline 28/29: Please rewrite the following sentence for better legibility!**
   **"This study is focusing on different GW breaking areas in the lower stratosphere along and the effects on the middle atmosphere highly depending on the position."**

We rewrote this part.
**P3L6-8**

**2. pg.2, line 29: who were concentrating ! who were focusing**

We replaced the word as suggested.
**P3L8**

**3. pg.3, line 2: shifting meridionally the EA/NP hotspot along its fixed longitude range! shifting meridionally the EA/NP hotspot keeping its longitude range fixed**

We changed this sentence as suggested.
**P3L16**

**4. pg.3, line 23: interval, in which ! interval in which**

We corrected this.
**P4L8**

**5. pg.4, line 3: refer to the mid ! referring to the mid**

We corrected this.
**P4L24**

**6. pg.4, line 17: reproduce ! reproduces**

We corrected this.
**P5L5**

**7. pg.5, line 2: observations but the ! observations, but the**

We corrected this.
**P5L10**

**8. pg.5, line 3: jet filtering some of ! jet filtering of some of**

We modified this sentence.
**P5L10**

**9. pg.7, line 4: is shifted toward ! that is shifted toward**

We corrected this.
**P8L21**

**10. pg.7, line 14: kmat ! km at**

We corrected this.
**P8L32**

**11. pg.8, line 14: west wind ! westerly wind**

We corrected this.
**P10L16**

**12. pg.11, line 10: more SPW 1 are excited, ! SPW 1 excitation is strengthened,**

We corrected this.
**P14L1**

**13. pg.12, line 2: fluxes and its ! fluxes and their**

We corrected this.
**P14L12**

**14. pg.13, line 3: atmopshere. ! atmosphere.**

We corrected this.
**P15L15**

**Comments of Reviewer 2:**

**The authors should include a table summarizing the different experiments.**

We agree with the referee. Owing to the huge amount of different experiments it would be better to summarize them in a table. This table includes the position as well as the different GW forcings in the respective region.
**P7**

**The reviewer found it a bit confusing that the Gaussian GW hotspot is shown in Figure 2 and the comparison between the 'used' simulations and the Gaussian is later presented in Figure 8. It is suggested to present this type of sensitivity in the experimental description section and later just refer to the 'main' model experiments.**

We agree that it might be a bit confusing. However, to evaluate the possible effect of a Gaussian distribution we first have to analyze at least the differences of the three-dimensional box H3 GW hotspot and the Ref simulation. We added more explanations to the description of Figure 2 and the new Table 1, and clearly stated that we will come back to the Gaussian-shaped hotspot later.
**P7L5-9, P8L1-3**

**Please clarify whether the altitudes shown in the Figures correspond to the pressure grid or whether the altitudes are computed from the geopotential and converted into a geometric altitude. This simplifies some comparisons to observations results.**

The altitude, which is shown or chosen in all figures is given in logarithmic pressure height. It is defined by:

$$z = -Hln\left(\frac{p}{p_0}\right)$$

z . . . . . . . . . . . . . . . . . . . . . . . . . . . . . . . . . . . . . . . . . . . . . . . . . . . . . . . . . . . . . .logarithmic pressure height
H = 7km . . . . . . . . . . . . . . . . . . . . . . . . . . . . . . . . . . . . . . . . . . . . . . . . . . . . . . . . . . . .scale height
p . . . . . . . . . . . . . . . . . . . . . . . . . . . . . . . . . . . . . . . . . . . . . . . . . . . . . . . . .pressure at specific level
$p_0$ = 1000hPa . . . . . . . . . . . . . . . . . . . . . . . . . . . . . . . . . . . . . . . . . . . . . . . . . . . reference pressure

The logarithmic pressure height corresponds to the geometrical height up to an altitude of about 110km with small deviations increasing within the altitude range. In 110 km the deviation is about 5 km. Depending on the thermospheric temperature the logarithmic pressure height can strongly differ from the geometrical height. Because we are just interested in altitudes up to 80 km, it can be assumed that the given altitude is close to the geometric height. We added an explanation in section 2.
**P3L27-31**

**Another comment of the reviewer concerns the implementation of the GW drag for the different latitudes. As the drag scales also with the atmospheric air mass that is affected by the drag, it might be mentioned that at higher latitudes essentially less drag is exerted to the atmosphere as the GW drag volume scales with latitude. Or with other words the atmospheric mass that is affected by the drag decreases with latitude. This might need some more discussion or should at least be mentioned in the interpretation of the results.**

The reviewer is right, the integrated drag scales with the cosine of latitude. This is, however, necessary, because otherwise the drag, both locally and as a zonal mean, would strongly increase with latitude. But in the governing equations such as e.g. the continuity equation the horizontal winds are scaled as well, so that this would result in a different experiment. We added a note on this in section 2.
**P8L4-13**

**Some Figures (3,4,6 and 7) need an improvement of the quality.**

We provided them in higher resolution.
**P9, P11, P12**

**Page 3: line 18: . . .vertical resolution 2842 km should be 2842 m**

We are using the '.' for the separation of decimal numbers.

[revised manuscript text omitted]

25   Typically, satellite observations show a characteristic structure of enhanced GW activity in the subtropical stratosphere that is caused by deep convection over Southeast Asia, America, Africa, or the Maritime Continent in the respective summer season (Jiang et al., 2004; Wright and Gille, 2011; Ern and Preusse, 2012).  Reliable estimates of GW drag from observations are generally difficult. Several methods have been established to derive the GW drag from satellite

30   (e.g. Ern et al., 2014, 2016), or from radar measurements (e.g. Reid and Vincent, 1987). However, uncertainties of these estimates are quite large.

From model studies there are indications that  GWs can already break in the lower stratosphere (LS)

35  (e.g. Plougonven et al., 2008; Constantino et al., 2015), which leads to an additional transfer of momentum and energy in this

region. In connection with high PW activity this is highly affecting the stability of the polar vortex and can cause a sudden stratospheric warming (SSW) (Albers and Birner, 2014). This effect has been also observed in satellite measurements showing enhanced GW drag before SSWs (Ern et al., 2016). Thus, an additional GW forcing may lead to a preconditioning of the polar vortex.

This study is focusing on  the role of the zonal position of localized GW breaking areas  and their effects on the middle atmosphere dynamics. It is motivated by the findings of Šácha et al. (2015) who were  focusing on the East Asian/North Pacific (EA/NP) region near Japan where they observed a GW hotspot being active during equinoxes and winter solstices. The GWs are orographically and convectively generated due to the topography directly at the coastline and the warm Kuroshio current. Šácha et al. (2015) analysed the local instabilities by calculating the Richardson number and by analysing reanalysis data and found that the GWs are breaking in this area. Based on these results they simulated the observed Asian GW breaking hotspot with a global circulation model (GCM) and analysed its effect in the middle atmosphere circulation (Šácha et al., 2016). According to previous publications e.g. by Smith (2003), Lieberman et al. (2013), or Matthias and Ern (2018),  Šácha et al. (2016) observed a forcing of additional stationary planetary waves (SPWs) due to a longitudinally variable GW drag. We are pursuing this idea by shifting meridionally the EA/NP hotspot  keeping its longitude range fixed to get information about its impact on the middle atmosphere at different latitudinal positions. Therefore, the EA/NP GW hotspot is our starting point, from which we are displacing the GW hotspot towards lower and higher latitudes in 5° steps. In section 2 of this paper, we provide a brief description of the GCM and detail the implementation of the GW hotspot within the GCM. In section 3 we describe and discuss the observed effects of the GW hotspots on the circulation of the middle atmosphere by analysing the stationary planetary wave activity and the propagation conditions. Finally, conclusions and outlook are presented in section 4.

**2 Numerical model experiments**

**2.1 Model description and set up**

To investigate the effect of localised GW breaking hotspots in the LS simulations have been performed using the Middle and Upper Atmosphere Model (MUAM, Pogoreltsev et al. (2007)). MUAM is a non-linear mechanistic 3D grid point model, which is an updated version of the global circulation model COMMA-LIM (Fröhlich et al., 2003a, 2007; Jacobi et al., 2006). The model extends in 56 layers up to an altitude of about 160 km in logarithmic pressure height $z = -H ln(p/p_0)$ with a constant scale height $H = 7$ km and the reference pressure $p_0 = 1000$ hPa. Depending on the temperature profile the used logarithmic pressure height can differ from the geometric height. However, at altitudes below 80 km, this difference is negligibly small. In 110 km the deviation is increasing up to 5 km, whereas the highest logarithmic pressure level of about 160 km may correspond to a geometrical height between 300 km and 400 km. In the lowermost 10 km, zonal mean temperatures are nudged to 2000-2010 mean monthly mean ERA Interim (Dee et al., 2011) zonal mean temperatures to correct the climatology of the troposphere, which is not included in detail in the model (Jacobi et al., 2015; Lilienthal et al., 2018). Furthermore, at 1000 hPa,

which defines the lower boundary of the model, SPWs of wavenumbers 1, 2 and 3 are forced, which are extracted from 2000-2010 mean ERA Interim monthly temperature and geopotential reanalysis data. The horizontal resolution of the model is $5°$ in latitude and $5.625°$ in longitude and the vertical resolution is $2.842\,\mathrm{km}$ . The model solves the primitive equations in flux form (e.g. Jakobs et al., 1986). MUAM includes parameterizations to simulate subgrid

5 processes such as GWs, absorption of solar radiation, or infrared cooling. The absorption of radiation is realized according to Strobel (1986). This parameterization is focused on the absorption processes due to trace gases such as $H_2O$ (absorber in the troposphere), $CO_2$ and $O_3$ (absorber in the stratosphere). Water vapor and ozone fields are prescribed. The heating rates are calculated by absorption bands representing the wavelength interval  in which these trace gases are absorbing the atmospheric radiation. Infrared emission of $CO_2$ is parameterized after Fomichev et al. (1998), and ozone infrared cooling in the $9.6\,\mu\mathrm{m}$

10 band is calculated after Fomichev and Shved (1985).

GWs are parameterized after an updated linear scheme (Lindzen, 1981; Jakobs et al., 1986) with multiple breaking levels (Fröhlich et al., 2003b; Jacobi et al., 2006). GW amplitudes are included at an altitude of $10\,\mathrm{km}$ as zonal mean with a global average of $1\,\mathrm{cms^{-1}}$ for the vertical velocity perturbation. This value is weighted by a prescribed zonal mean GW amplitude distribution based on  $E_{pot}$ data obtained from GPS radio occultation measurements (Šácha et al., 2015; Lilien-

15 thal et al., 2017). Although the $E_{pot}$ data still contain Kelvin waves and other possible wave structures with short vertical wavelengths, which may introduce biases, the GW amplitude distribution is more realistic compared to the hyperbolic tangent function of the latitude, which was used in earlier experiments (Jacobi et al., 2006) and leads to an improvement of the zonal mean GW climatology. It shows maximum GW amplitudes (not shown here) at the equator (convectively generated GWs) and at midlatitudes (orographically induced GWs). At each grid point 48 waves are induced propagating in eight different direc-

20 tions with six different phase speeds ranging from 5 to $30\,\mathrm{ms^{-1}}$.

In this configuration based on January decadal mean (2000-2010) ERA Interim reanalysis data we create a reference simulation with a spin-up period of 270 days, in which the mean circulation is built up and different waves like planetary waves (PWs) and tides are generated. The declination and the ozone and carbon dioxide concentration are fixed to avoid further non-zonal structures being induced besides to the enhanced GW forcing. The declination corresponds to January 15 ( referring to

25 the mid of the month) and the ozone and carbon dioxide data are taken from the year 2005 (refer to the mid of the decade). For the analysis, a time interval of 120 days with a temporal resolution of 2 hours after the spin-up period was modeled. Šácha et al. (2016) have already  analysed the effect of the Asian hotspot with MUAM by performing a sensitivity study with regard to the strength of the GW forcing in the stratosphere.

Their analysis time period was much shorter and the declination of the sun was different. They also nudged the model zonal

30 mean temperature up to $30\,\mathrm{km}$.  In this regard, our experimental set-up might be considered superior to their simulations, especially, because the nudging does not interfere with the implemented GW forcing  in this new configuration . We refer to this reference simulation as the 'Ref' simulation.

The state of the middle atmosphere of the Ref simulation can be seen in Fig. 1, which shows the January zonal mean zonal (a) and meridional wind (b), the temperature (c), zonal GW flux (d), the zonal wind acceleration due to breaking GWs (e), and the

35 SPW 1 amplitude extracted from the zonal wind (f) as latitude-height plots. Each parameter is presented up to an altitude of

[Figure]

**Figure 1.** January zonal and monthly mean of the (a) zonal wind (ms$^{-1}$), (b) meridional wind (ms$^{-1}$), (c) temperature (K), (d) zonal GW fluxes (m$^2$s$^{-2}$), (e) zonal wind acceleration through breaking GWs (ms$^{-1}$day$^{-1}$) and (f) SPW 1 amplitude (ms$^{-1}$) extracted from the zonal wind of the reference simulation.

120 km for the winter and summer hemisphere. The zonal wind in Fig. 1(a) generally reproduces reference climatologies like CIRA-86 (Fleming et al., 1988) or URAP (Swinbank and Ortland, 2003), but the winter mesospheric jet is overestimated by about 10-20 ms$^{-1}$. The meridional circulation (Fig. 1(b)) extending from the summer to the winter mesopause has a maximum of 6 ms$^{-1}$ at about 80 km, which is well reproducing predictions by climatologies (Portnyagin et al., 2004; Jacobi et al., 2009).

5 Temperature (Fig. 1(c)) generally  reproduces climatology values. The GW fluxes (Fig. 1(d)) maximize at about 80 km, with a maximum of slightly above -4 m$^2$s$^{-2}$ on the northern hemisphere (NH) and 2 m$^2$s$^{-2}$ on the southern hemisphere (SH). The corresponding zonal GW drag maximizes at the same altitude with about -60 ms$^{-1}$day$^{-1}$ (40 ms$^{-1}$day$^{-1}$) in the NH (SH) and is westward (eastward) directed. The SPW 1 amplitude (Fig. 1(f)) extracted from the zonal wind shows maximum values at the border of the mesospheric jet maximum northward of 30°N between 50 and 60 km and in the polar region. This

10 fits quite well to observations, but the amplitudes are slightly underestimated due to the overestimated mesospheric jet, filtering some of the SPWs (Xiao et al., 2009).

**2.2 Experiment description**

In a first experiment, we reproduced the experiment of Šácha et al. (2016) to check if we still get similar results with the slightly modified setup. To represent the Asian GW breaking hotspot in the model we enhanced the GW drag after model day 270, i.e.

15 after the spin up, and run the model for 120 days as in the Ref simulation.

Therefore, the zonal (GWD$_u$) and meridional (GWD$_v$) GW drag and the heating due to breaking GWs (GWD$_T$) are modified in the specific region of the observed GW breaking hotspot. In principle, the response to the GW drag would in turn alter the

[Figure]

**Figure 2.** Zonal GW drag ($\mathrm{ms^{-1}day^{-1}}$) at 26.9 km for the reference (left) and the H3 hotspot simulation as box (middle) and as Gaussian distribution (right) for the last 30 days of analysis. Note the different scaling on the left panel.

GW propagation and breaking conditions and thus, the GW drag and its distribution. To avoid those feedback mechanisms, the GW parameterization scheme is turned of during the experiments and the model is fed with the GW drag field from the Ref simulation. Only, in the GW hotspot region the GW drag is modified as shown in Tab. 1. We intend to only analyse the steady state impact of the local GW forcing being not influenced by nonlinear effects. Like in Šácha et al. (2016) we located

5 the GW breaking hotspot between 37.5°N-62.5°N and 118.1°E-174.3°E in an altitude range between 18 and 30 km. Note that the geographic positions refer to the model grid points so that at a latitudinal 5° grid the meridional size of the modeled hotspot is 30°. To avoid a total breakdown of the polar vortex and a fundamental change in middle atmosphere dynamics, which was already forced in the study by Šácha et al. (2016), we chose the more moderate case of -10 $\mathrm{ms^{-1}day^{-1}}$ for $\mathrm{GWD}_u$, -0.1 $\mathrm{ms^{-1}day^{-1}}$ for $\mathrm{GWD}_v$ and a warming of 0.05 $\mathrm{Kday^{-1}}$ for $\mathrm{GWD}_T$. We refer to this simulation as the H3 simulation, as

10 will be described later. The distribution of the $\mathrm{GWD}_u$ of the Ref and the H3 simulation can be seen in the left and middle panel of Fig. 2 at an altitude of about 27 km for the last 30 days of analysis. We are mainly concentrating on the last 30 days of analysis because we are focusing on quasi steady states and are not interested in short term variabilities. The $\mathrm{GWD}_u$ of the Ref simulation is varying between -0.025 and +0.02 $\mathrm{ms^{-1}day^{-1}}$ in the region of the GW hotspots (27.5°N-87.5°N, 118.1°E-174.3°E, 18-30 km). Thus, the maximum value of the H3 simulation ($\mathrm{GWD}_u$= -10 $\mathrm{ms^{-1}day^{-1}}$

15 in the hotspot) is 500 times larger than the maximum westward (negative) value of the Ref simulation.

The H3 mean value (mean $\mathrm{GWD}_u$: -10 $\mathrm{ms^{-1}day^{-1}}$) is roughly 3300 larger than the one of the Ref simulation (mean $\mathrm{GWD}_u$: 0.003 $\mathrm{ms^{-1}day^{-1}}$) within the region of the EA/NP hotspot. These maximum values of the $\mathrm{GWD}_u$ as well as those of the $\mathrm{GWD}_v$ and the $\mathrm{GWD}_T$ are summarized in Tab. 1 for the Ref and the GW hotspot simulations. In spite of the huge difference compared to the Ref simulation, the zonal GW forcing is moderate in terms of what is estimated from observations (40 $\mathrm{ms^{-1}day^{-1}}$ and

20 more) and from GW parameterizations in this region (Šácha et al., 2018). Concerning the meridional GW drag and the heating due to breaking GWs the maximum (mean) value of the H3 simulation is only 5 (100) times larger than the one of the Ref simulation (not shown here). To investigate possible effects with regard to the position of the GW hotspot we performed a sensitivity study. For this, we kept the longitude (118.1°E-174.3°E) and altitude (18-30 km) range as well as the zonal extent of 25° fixed, but varied the observed GW hotspot in 5° steps from 27.5°N-52.5°N (simulation H1) to 62.5°N-87.5°N (simulation

| Simulation | Abbreviation | Region | Min./Max. GWD$_u$ (ms$^{-1}$day$^{-1}$) | Min./Max. GWD$_v$ (ms$^{-1}$day$^{-1}$) | Min./Max. GWD$_T$ (Kday$^{-1}$) |
|---|---|---|---|---|---|
| Reference | Ref | | -0.025/0.02 | -0.025/0.01 | -0.0006/0.01 |
| Hotspot | H1 | 27.5-52.5°N | -10 | -0.1 | 0.05 |
| | H2 | 32.5-57.5°N | | | |
| | H3 | 37.5-62.5°N | | | |
| | H4 | 42.5-67.5°N | | | |
| | H5 | 47.5-72.5°N | | | |
| | H6 | 52.5-77.5°N | | | |
| | H7 | 57.5-82.5°N | | | |
| | H8 | 62.5-87.5°N | | | |
| | | 118.1-174.3°E 18-30 km | | | |
| Gauss | Gauss | | -13.1 | -0.13 | 0.065 |

**Table 1.** Overview of the mean and maximum values of the zonal and meridional GW drag and heating by GWs for the reference and hotspot simulations as three dimensional box and as Gaussian distribution. The mean and maximum values refer to the region (118.1-174.3°E, 18-30 km and the respective latitude range) of the hotspots.

H8), while labeling the experiments inbetween by H2 through H7 (see Tab. 1).

To analyze possible effects of the sharp transition zone between the unchanged and enhanced GW drag, additional simulations with a smoothed GW forcing were performed, by using a 3D Gaussian function, with a standard deviation of 10°, 22.5°, and 5.684 km in the zonal, meridional, and vertical direction, respectively. To get the same integral forcing as in the H1-H8 simulations, the size or the intensity of the local GW forcing as Gaussian distribution needs to be adjusted. For our experiments, we mainly increased the strength of the local GW forcing and only slightly increased the size. The maximum values for the GWD$_u$, GWD$_v$, and GWD$_T$ forcing as 3D Gaussian distribution have been chosen as -13 ms$^{-1}$day$^{-1}$, -0.13 ms$^{-1}$day$^{-1}$, and 0.065 Kday$^{-1}$ (see Tab. 1), respectively. For comparison, we are just focussing The 3D Gaussian distribution for the H3 GW hotspot can be seen in the right panel of Fig. 2. In this paper we mainly concentrate on the 3D GW hotspots shaped like a

box, when we analyse the effects on the middle atmosphere dynamics. For comparison regarding the shape of the artificial GW forcing, we just focus on the H3  GW hotspot with Gaussian smoothed boundaries, when we discuss the SPW modulation in section 3.2, which may be affected by the GW hotspots with sharp boundaries.

By comparing the size of the GW hotspots it is obvious (can be seen in Fig. 5) that the area of the enhanced GW drag, which scales with the cosine of the latitude, is decreasing for increasing latitude. But scaling the GW drag with latitude would lead to a much larger zonal mean GW drag at high latitudes and resulting changes in the circulation. Also the horizontal winds, which are affected by resulting nonlinear interactions are scaled in the model equations. In the current approach we are conserving the ratio of enhanced and unchanged GW drag values within the respective latitudinal belt, which is more meaningful. Also, the horizontal wavelength of PWs is smaller the closer to the pole they are, so that the ratio of the width of the GW forcing and the horizontal wavelength of the PWs remains the same for the respective latitudinal belt. In the following we will show that the spatial shape as well as the spatial size of the local GW forcing is not the most decisive factor, when we are comparing the 3D Gaussian distribution with the 3D GW forcing shaped as a box. Thus, GW hotspots having all the same size may lead to comparable results.

[revised manuscript text omitted]

To sum up the influence on the polar vortex, in section 3.1 we have observed a slight warming of the lower stratosphere and the decreasing west wind at middle to high latitudes, which indicates a weakening of the polar vortex as a consequence of the GW drag enhancement. Thus, the stability of the polar vortex is not only depending on the PW activity but rather on the interplay or nonlinear interaction of GW and PW forcings. The anomalous SPW forcing and suppression of SPW propagation show that the GW drag can play an important role for the preconditioning of the polar vortex (see next section).

[revised manuscript text omitted]

5   Based on these results of the sensitivity study we have seen that a local GW forcing can lead to a weakening (warming of the lower stratosphere and slight displacement) of the polar vortex at high latitudes, which is highly depending on the strength (Šácha et al., 2016) as well as on the zonal distribution of the forcing (this study). Usually, it is assumed that the preconditioning of the polar vortex is mainly driven by enhanced PW activity (Labitzke, 1981). But there are also several indications based on satellite observations (Ern et al., 2016) and reanalysis data (Albers and Birner, 2014) showing that the GW drag and the

10  absolute GW momentum flux is enhanced (reduced) in the stratosphere right before (after) SSWs. Albers and Birner (2014) have analyzed the total wave forcing from Japanese Meteorological Agency and Central Research Institute of Electrical Power Industry 25-year Reanalysis (JRA-25) data before SSW events and found that up to 70% of the total drag is induced by orographic GWs. Ern et al. (2016) directly derived the GW drag and absolute momentum fluxes from HIRDLS and SABER temperatures and found out that both parameters are enhanced before and around the central day of a SSW (strong polar jet)

15  and reduced, when the zonal wind is weak (after SSW). Because we kept the GW drag forcing constant during the whole experiments we cannot evaluate nonlinear effects, which would possibly reduce the GW drag connected with the displacement of the polar vortex. Furthermore, we have a fixed GW source distribution so that no additional GWs are generated owing to changes in the tropospheric circulation. But on the basis of the zonal and meridional GW flux, which is changing according to the propagation condtions, we have analyzed the absolute horizontal GW momentum flux (not shown here). In this case, we

20  observe as well a reduction of the GW flux, when the zonal mean zonal wind is decreasing at high latitudes. However, this effect is not very pronounced in our experiments because the zonal mean zonal wind differences are much smaller than during a real SSW event. We only observe zonal mean zonal wind differences of about $-10\,\mathrm{ms}^{-1}$, which do not lead to a reversal of the zonal mean zonal wind, and thus, to significant background changes, which may strongly influence the GW propagation.

[revised manuscript text omitted]

---

## Author Response (AR2)

Dear Referee 1,

thank you for your helpful comments. We have included them all into our revised paper.

**pg.3, line 4: this term is more often used:**
**global circulation model (GCM) -> general circulation model (GCM)**

Done.

**pg.6, line 4: is turned of -> is turned off**

Done.

**pg.12, line 14: The SPW 2 amplitude it can be seen -> From the SPW 2 amplitude it can be seen**

Done.

**pg.17, line 24: on the middle atmosphere. -> of the middle atmosphere.**

Done.

**pg.17, line 26:**
**stratosphere and slight displacement) of the polar vortex**
**->**
**stratosphere) and slight displacement of the polar vortex**

Done.

[revised manuscript text omitted]

10 usually observed in the middle atmosphere. Also GWs being faster than the background wind are able to propagate but they are mostly filtered out by the strong polar-night jet at the latest, when $c$ becomes equal to $u$. In the mesosphere GWs, which are propagating in the opposite direction to $u$ saturate and deposit their momentum. For this reason the wind reverses in the upper mesosphere/lower thermosphere (MLT) (Lindzen, 1981; Holton, 1982). The transfer of energy and momentum by breaking GWs is also called GW drag.

15 Owing to the variety of sources GWs have a large spatial and temporal variability. To capture the global distribution of GWs, the potential energy ($E_{pot}$), momentum flux (MF), or stability indicators (Pišoft et al., 2018) can be estimated by using satellite data (Ern et al., 2004; Fröhlich et al., 2007; Hoffmann et al., 2013; Schmidt et al., 2016). These numerous observational studies highlight a number of different local GW hotspots, which are mainly generated by orography and convection. The most common GW hotspots are the orographically induced GW hotspots near the Alps (Hierro et al., 2018), the Andes (Llamedo et al.,

20 2009; Alexander et al., 2010; Lilienthal et al., 2017), the Antarctic Peninsula (Moffat-Griffin et al., 2010), the Himalayas (Kumar et al., 2012), the Mongolian Plateau (White et al., 2018), the Rocky Mountains (Lilly et al., 1982) and in the Scandinavian region (Kirkwood et al., 2010). Typically, satellite observations show a characteristic structure of enhanced GW activity in the subtropical stratosphere that is caused by deep convection over Southeast Asia, America, Africa, or the Maritime Continent in the respective summer season (Jiang et al., 2004; Wright and Gille, 2011; Ern and Preusse, 2012). Reliable estimates of GW

25 drag from observations are generally difficult. Several methods have been established to derive the GW drag from satellite (e.g. Ern et al., 2014, 2016), or from radar measurements (e.g. Reid and Vincent, 1987). However, uncertainties of these estimates are quite large.

From model studies there are indications that GWs can already break in the lower stratosphere (LS) (e.g. Plougonven et al., 2008; Constantino et al., 2015), which leads to an additional transfer of momentum and energy in this region. In connection

30 with high PW activity this is highly affecting the stability of the polar vortex and can cause a sudden stratospheric warming (SSW) (Albers and Birner, 2014). This effect has been also observed in satellite measurements showing enhanced GW drag before SSWs (Ern et al., 2016). Thus, an additional GW forcing may lead to a preconditioning of the polar vortex.

This study is focusing on the role of the zonal position of localized GW breaking areas and their effects on the middle atmosphere dynamics. It is motivated by the findings of Šácha et al. (2015) who were focusing on the East Asian/North Pacific

35 (EA/NP) region near Japan where they observed a GW hotspot being active during equinoxes and winter solstices. The GWs

are orographically and convectively generated due to the topography directly at the coastline and the warm Kuroshio current. Šácha et al. (2015) analysed the local instabilities by calculating the Richardson number and by analysing reanalysis data and found that the GWs are breaking in this area. Based on these results they simulated the observed Asian GW breaking hotspot with a  general circulation model (GCM) and analysed its effect in the middle atmosphere circulation (Šácha et al.,

5   2016). According to previous publications e.g. by Smith (2003), Lieberman et al. (2013), or Matthias and Ern (2018), Šácha et al. (2016) observed a forcing of additional stationary planetary waves (SPWs) due to a longitudinally variable GW drag. We are pursuing this idea by shifting meridionally the EA/NP hotspot keeping its longitude range fixed to get information about its impact on the middle atmosphere at different latitudinal positions. Therefore, the EA/NP GW hotspot is our starting point, from which we are displacing the GW hotspot towards lower and higher latitudes in $5°$ steps. In section 2 of this paper, we

10   provide a brief description of the GCM and detail the implementation of the GW hotspot within the GCM. In section 3 we describe and discuss the observed effects of the GW hotspots on the circulation of the middle atmosphere by analysing the stationary planetary wave activity and the propagation conditions. Finally, conclusions and outlook are presented in section 4.

**2   Numerical model experiments**

**2.1   Model description and set up**

15   To investigate the effect of localised GW breaking hotspots in the LS simulations have been performed using the Middle and Upper Atmosphere Model (MUAM, Pogoreltsev et al. (2007)). MUAM is a non-linear mechanistic 3D grid point model, which is an updated version of the  general circulation model COMMA-LIM (Fröhlich et al., 2003a, 2007; Jacobi et al., 2006). The model extends in 56 layers up to an altitude of about $160\,\mathrm{km}$ in logarithmic pressure height $z = -Hln(p/p_0)$ with a constant scale height $H = 7\,\mathrm{km}$ and the reference pressure $p_0 = 1000\,\mathrm{hPa}$. Depending on the temperature profile the used log-

20   arithmic pressure height can differ from the geometric height. However, at altitudes below $80\,\mathrm{km}$, this difference is negligibly small. In $110\,\mathrm{km}$ the deviation is increasing up to $5\,\mathrm{km}$, whereas the highest logarithmic pressure level of about $160\,\mathrm{km}$ may correspond to a geometrical height between $300\,\mathrm{km}$ and $400\,\mathrm{km}$. In the lowermost $10\,\mathrm{km}$, zonal mean temperatures are nudged to 2000-2010 mean monthly mean ERA Interim (Dee et al., 2011) zonal mean temperatures to correct the climatology of the troposphere, which is not included in detail in the model (Jacobi et al., 2015; Lilienthal et al., 2018). Furthermore, at $1000\,\mathrm{hPa}$,

[revised manuscript text omitted]

15 Ref simulation is varying between -0.025 and +0.02 ms$^{-1}$day$^{-1}$ in the region of the GW hotspots (27.5°N-87.5°N, 118.1°E-174.3°E, 18-30 km). Thus, the maximum value of the H3 simulation ($\text{GWD}_u$= -10 ms$^{-1}$day$^{-1}$ in the hotspot) is 500 times larger than the maximum westward (negative) value of the Ref simulation.

| Simulation | Abbreviation | Region | Min./Max. $\text{GWD}_u$ (ms$^{-1}$day$^{-1}$) | Min./Max. $\text{GWD}_v$ (ms$^{-1}$day$^{-1}$) | Min./Max. $\text{GWD}_T$ (Kday$^{-1}$) |
|---|---|---|---|---|---|
| Reference | Ref | | -0.025/0.02 | -0.025/0.01 | -0.0006/0.01 |
| Hotspot | H1 | 27.5-52.5°N | -10 | -0.1 | 0.05 |
| | H2 | 32.5-57.5°N | | | |
| | H3 | 37.5-62.5°N | | | |
| | H4 | 42.5-67.5°N | | | |
| | H5 | 47.5-72.5°N | | | |
| | H6 | 52.5-77.5°N | | | |
| | H7 | 57.5-82.5°N | | | |
| | H8 | 62.5-87.5°N | | | |
| | | 118.1-174.3°E | | | |
| | | 18-30 km | | | |
| Gauss | Gauss | | -13.1 | -0.13 | 0.065 |

**Table 1.** Overview of the mean and maximum values of the zonal and meridional GW drag and heating by GWs for the reference and hotspot simulations as three dimensional box and as Gaussian distribution. The mean and maximum values refer to the region (118.1-174.3°E, 18-30 km and the respective latitude range) of the hotspots.

The H3 mean value (mean $GWD_u$: -10 ms$^{-1}$day$^{-1}$) is roughly 3300 larger than the one of the Ref simulation (mean $GWD_u$: 0.003 ms$^{-1}$day$^{-1}$) within the region of the EA/NP hotspot. These maximum values of the $GWD_u$ as well as those of the $GWD_v$ and the $GWD_T$ are summarized in Tab. 1 for the Ref and the GW hotspot simulations. In spite of the huge difference compared to the Ref simulation, the zonal GW forcing is moderate in terms of what is estimated from observations (40 ms$^{-1}$day$^{-1}$ and more) and from GW parameterizations in this region (Šácha et al., 2018). Concerning the meridional GW drag and the heating due to breaking GWs the maximum (mean) value of the H3 simulation is only 5 (100) times larger than the one of the Ref simulation (not shown here). To investigate possible effects with regard to the position of the GW hotspot we performed a sensitivity study. For this, we kept the longitude (118.1°E-174.3°E) and altitude (18-30 km) range as well as the zonal extent of 25° fixed, but varied the observed GW hotspot in 5° steps from 27.5°N-52.5°N (simulation H1) to 62.5°N-87.5°N (simulation H8), while labeling the experiments inbetween by H2 through H7 (see Tab. 1).

To analyze possible effects of the sharp transition zone between the unchanged and enhanced GW drag, additional simulations with a smoothed GW forcing were performed, by using a 3D Gaussian function, with a standard deviation of 10°, 22.5°, and 5.684 km in the zonal, meridional, and vertical direction, respectively. To get the same integral forcing as in the H1-H8 simulations, the size or the intensity of the local GW forcing as Gaussian distribution needs to be adjusted. For our experiments, we mainly increased the strength of the local GW forcing and only slightly increased the size. The maximum values for the $GWD_u$, $GWD_v$, and $GWD_T$ forcing as 3D Gaussian distribution have been chosen as -13 ms$^{-1}$day$^{-1}$, -0.13 ms$^{-1}$day$^{-1}$, and 0.065 Kday$^{-1}$ (see Tab. 1), respectively. The 3D Gaussian distribution for the H3 GW hotspot can be seen in the right panel of Fig. 2. In this paper we mainly concentrate on the 3D GW hotspots shaped like a box, when we analyse the effects on the middle atmosphere dynamics. For comparison regarding the shape of the artificial GW forcing, we just focus on the H3 GW hotspot with Gaussian smoothed boundaries, when we discuss the SPW modulation in section 3.2, which may be affected by the GW hotspots with sharp boundaries.

By comparing the size of the GW hotspots it is obvious (can be seen in Fig. 5) that the area of the enhanced GW drag, which scales with the cosine of the latitude, is decreasing for increasing latitude. But scaling the GW drag with latitude would lead to a much larger zonal mean GW drag at high latitudes and resulting changes in the circulation. Also the horizontal winds, which are affected by resulting nonlinear interactions are scaled in the model equations. In the current approach we are conserving the ratio of enhanced and unchanged GW drag values within the respective latitudinal belt, which is more meaningful. Also, the horizontal wavelength of PWs is smaller the closer to the pole they are, so that the ratio of the width of the GW forcing and the horizontal wavelength of the PWs remains the same for the respective latitudinal belt. In the following we will show that the spatial shape as well as the spatial size of the local GW forcing is not the most decisive factor, when we are comparing the 3D Gaussian distribution with the 3D GW forcing shaped as a box. Thus, GW hotspots having all the same size may lead to comparable results.

[revised manuscript text omitted]

To sum up the influence on the polar vortex, in section 3.1 we have observed a slight warming of the lower stratosphere and the decreasing west wind at middle to high latitudes, which indicates a weakening of the polar vortex as a consequence of the GW drag enhancement. Thus, the stability of the polar vortex is not only depending on the PW activity but rather on the interplay or nonlinear interaction of GW and PW forcings. The anomalous SPW forcing and suppression of SPW propagation show that
20  the GW drag can play an important role for the preconditioning of the polar vortex (see next section).

[revised manuscript text omitted]

25  Based on these results of the sensitivity study we have seen that a local GW forcing can lead to a weakening (warming of the lower stratosphere) and slight displacement ) of the polar vortex at high latitudes, which is highly depending on the strength (Šácha et al., 2016) as well as on the zonal distribution of the forcing (this study). Usually, it is assumed that the preconditioning of the polar vortex is mainly driven by enhanced PW activity (Labitzke, 1981). But there are also several indications based on satellite observations (Ern et al., 2016) and reanalysis data (Albers and Birner, 2014) showing that the GW drag and the

30  absolute GW momentum flux is enhanced (reduced) in the stratosphere right before (after) SSWs. Albers and Birner (2014) have analyzed the total wave forcing from Japanese Meteorological Agency and Central Research Institute of Electrical Power Industry 25-year Reanalysis (JRA-25) data before SSW events and found that up to 70% of the total drag is induced by orographic GWs. Ern et al. (2016) directly derived the GW drag and absolute momentum fluxes from HIRDLS and SABER temperatures and found out that both parameters are enhanced before and around the central day of a SSW (strong polar jet)

and reduced, when the zonal wind is weak (after SSW). Because we kept the GW drag forcing constant during the whole experiments we cannot evaluate nonlinear effects, which would possibly reduce the GW drag connected with the displacement of the polar vortex. Furthermore, we have a fixed GW source distribution so that no additional GWs are generated owing to changes in the tropospheric circulation. But on the basis of the zonal and meridional GW flux, which is changing according to the propagation condtions, we have analyzed the absolute horizontal GW momentum flux (not shown here). In this case, we observe as well a reduction of the GW flux, when the zonal mean zonal wind is decreasing at high latitudes. However, this effect is not very pronounced in our experiments because the zonal mean zonal wind differences are much smaller than during a real SSW event. We only observe zonal mean zonal wind differences of about -10 ms$^{-1}$, which do not lead to a reversal of the zonal mean zonal wind, and thus, to significant background changes, which may strongly influence the GW propagation.

[revised manuscript text omitted]